# Decadal sea-level variability in the Australasian Mediterranean Sea

Patrick Wagner[1] and Claus W. Böning[1]

[1]GEOMAR Helmholtz Centre for Ocean Research Kiel, Kiel Germany

**Correspondence:** Patrick Wagner (pwagner@geomar.de)

**Abstract.**

Strong regional sea-level trends, mainly related to basin-wide wind stress anomalies, have been observed in the western tropical Pacific over the last three decades. Analyses of regional sea level in the densely populated regions of the neighbouring Australasian Mediterranean Sea (AMS; also called Tropical Asian Seas) are hindered by its complex topography and respective studies are sparse. We used a series of global, eddy-permitting ocean models, including a high-resolution configuration that resolves the AMS with $\frac{1}{20}°$ horizontal resolution, forced by a comprehensive atmospheric forcing product over 1958–2016, to characterize the patterns and magnitude of decadal sea-level variability in the AMS. The nature of this variability is elucidated further by sensitivity experiments with interannual variability restricted to either the momentum or buoyancy fluxes, building on an experiment employing a repeated-year forcing without interannual variability in all forcing components. Our results suggest that decadal fluctuations of El Niño-Souther Oscillation (ENSO) account for over 80% of the variability in all deep basins of the region, except for the central South China Sea (SCS). Changes related to the Pacific Decadal Oscillation (PDO) are most pronounced in the shallow Arafura and Timor seas and in the central SCS. On average, buoyancy fluxes account for less than 10% of decadal SSH variability, but this ratio is highly variable over time and can reach values of up to 50%. In particular, our results suggest that buoyancy flux forcing amplifies the dominant wind-stress-driven anomalies related to ENSO cycles. Intrinsic variability is mostly negligible except in the SCS, where it accounts for 25% of the total decadal SSH variability.

## 1 Introduction

Between the early 1990s and mid 2010s sea-level trends of up to 10 mm yr$^{-1}$, which exceeds three times the global mean rate, were observed in the western tropical Pacific and found to be related to an intensification of the Pacific trade wind regime (Timmermann et al., 2010; Merrifield, 2011; Merrifield and Maltrud, 2011; McGregor et al., 2012; Merrifield et al., 2012; Moon and Song, 2013; Moon et al., 2013; Qiu and Chen, 2012). This strong sea-level rise had severe consequences for the population of the low-lying islands in the region (Becker et al., 2012), whose inhabitants are regularly referred to as the first climate refugees. Sea-level changes in the western tropical Pacific received a lot of scientific attention. Despite the exposed position of many densely populated, coastal communities, this is not true for the neighbouring areas of the Australasian Mediterranean Sea (AMS; see Fig. 1 for an overview map of the study domain). For the period 1993–2009 strong trends of up to 9 mm yr$^{-1}$ have been observed in the southwestern part of the region, but trends are not uniform across the many small seas of the region

(Strassburg et al., 2015) and studies on regional sea-level variability in the region are sparse.

The best studied area in the region in terms of sea-level is the South China Sea (SCS). It is the largest marginal sea of the region. The central and northeastern deep parts of the basin show the largest amplitudes of interannual and decadal variability (Cheng and Qi, 2007), with trends of up to 8 mm yr$^{-1}$ for the period 1993-2012 (Cheng et al., 2016). There is a broad consensus that El Niño-Souther Oscillation (ENSO) has a strong impact on interannual sea-level variability (Cheng and Qi, 2007; Cheng et al., 2016; Rong et al., 2007; Wu and Chang, 2005; Fang et al., 2006; Peng et al., 2013). Subsurface heat content anomalies are advected into the basin via the Luzon Strait (Cheng and Qi, 2007; Rong et al., 2007) and coastal Kelvin Waves advect anomalies, related to ENSO and possibly the Pacific Decadal Oscillation (PDO), from the tropical Pacific clockwise around the Philippines into the SCS (Liu et al., 2011; Zhuang et al., 2013; Cheng et al., 2016). Local wind stress curl, also related to ENSO, might amplify these sea-level anomalies in the central SCS (Cheng et al., 2016). In contrast, Kleinherenbrink et al. (2017) did not find a strong impact of tropical wind stress variability on sea-level changes in the SCS.

Sea-level variability in the southeastern part of the AMS, i.e. between Borneo and Australia, received much less attention. However, the area has been of great interest since it represents the only low-latitude connection of ocean basins (Sprintall et al., 2014) and acts as an upper pathway of the global overturning circulation (Gordon, 1986). The main circulation feature is the Indonesian Throughflow (ITF) that transports heat and freshwater from the Pacific into the Indian Ocean (Gordon, 2005). The ITF variability on interannual time scales is mostly a response to variations of the Pacific trade winds and governed by large-scale climate modes. In particular, ENSO has a strong effect on ITF transport on interannual time scales (Meyers, 1996; Wijffels and Meyers, 2004; Liu et al., 2015). Decadal ITF variability is also linked to shifts of the Pacific trade wind regime (Wainwright et al., 2008; Liu et al., 2010; Zhuang et al., 2013; Feng et al., 2011, 2015), but other mechanism, like deep Pacific upwelling (Feng et al., 2017), could also play a role. However, given the impact of Pacific easterlies on ITF variability, it is reasonable to assume that sea level along the ITF pathways is also remotely controlled by the Pacific trade wind regime. Indeed, sea-level variability along the Australian west coast is closely linked to variability in the western tropical Pacific (Feng et al., 2004; Lee and McPhaden, 2008; Merrifield et al., 2012). This requires a way for signals to cross the AMS. The theory of an equatorial waveguide, that allows remote forcing of variability in the throughflow region, was already proposed in 1994 (Clarke and Liu, 1994). Wijffels and Meyers (2004) showed, based on XBT observations from 1984 to 2001, that sea-level anomalies propagate from the Pacific Ocean along the Papuan and Australian shelf break into the Indian Ocean and that interannual variability is driven by remote wind forcing from the Pacific.

In a more detailed analysis, Kleinherenbrink et al. (2017) separated observed sea-level changes between 2005 and 2012 into mass and steric components. They found mass changes to be relevant in the shelf regions, where a shallow water column is unable to expand due to density changes. Furthermore, they reported that the deep basins of the Banda and Celeb seas show large steric sea-level variability on interannual time scales which appear to be linked to wind stress anomalies over the tropical Pacific. Using the Dipole Mode Index (DMI; Saji et al. 1999) they also found an impact of the Indian Ocean that is however

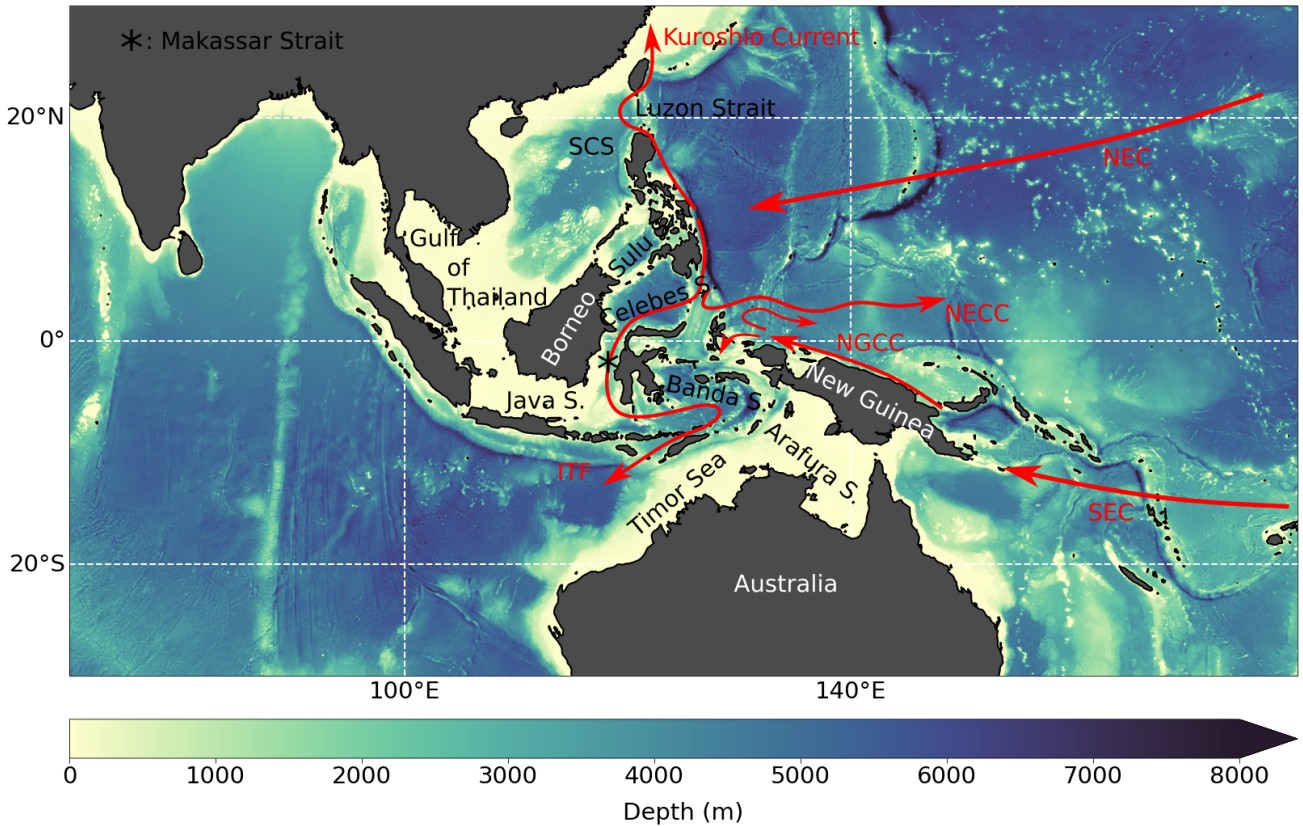

**Figure 1.** *Overview map of study domain, depicting the marginal seas and schematic currents discussed in the text. Colourshading indicates depth. Marginal seas, islands, continents and schematic currents that are mentioned in the text are also marked. SCS: South China Sea, NEC: North Equatorial Current, SEC: South Equatorial Current, NECC: North Equatorial Counter Current.*

limited to the Timor Sea. Strassburg et al. (2015) could also link observed and reconstructed decadal sea-level trends between 1950 and 2009 to PDO-related wind stress changes over the tropical Pacific.

65   Comparatively little is known about the importance of local or remote buoyancy fluxes that has been proven to be relevant for sea level in other parts of the world's ocean, including the tropical Pacific (e.g. Piecuch and Ponte 2011; Piecuch et al. 2019; Wagner et al. 2021). Cheng and Qi (2007) suggested an influence of ocean–atmosphere freshwater fluxes on halosteric sea level in the SCS. In particular, the observed decrease between 2001 and 2005 was found to coincide with a salinification trend due to reduced freshwater fluxes. Rong et al. (2007) pointed out that local precipitation anomalies are the primary control
70   of the SCS-mass budget and thereby also influences sea level.

Using ocean and climate models to analyse sea level in this region is hindered by its complex topography that forms many small seas and narrow straits. For example, the Labani channel, the deep part of the Makassar strait and the main pathway of the ITF, is only about 40 km broad, which is barely resolved in an 1/4° ocean model, let alone in even coarser climate models. We therefore used a series of ocean model experiments that consists of two hindcast simulations that subject to realistic atmospheric forcing covering the last six decades and providing horizontal resolution of 1/4°and 1/20°, supplemented by a series of sensitivity experiments that allow an individual application of momentum and buoyancy flux forcing to the underlying ocean. The aim of this study is to examine the sea-level variability in the AMS in high-resolution ocean model simulations subject to atmospheric forcing over the last six decades. In particular, the objectives are:

– To elucidate the patterns and nature of the regional sea-level variability on decadal to multi-decadal time scales, including the impacts of the low-frequency ENSO and PDO changes which have been identified in previous studies as major drivers of sea-level variability on interannual to decadal time scales.

– To examine the possible contribution of intrinsic variability, internally generated by ocean dynamics, to the total sea-level variability in the region, and to determine the relative roles of momentum and buoyancy fluxes to the forced variability.

– To analyse the individual contributions of temperature and salinity changes to steric sea-level variability.

Concerning the notation of the marginal seas between Australia and Asia, some deviating conventions have been used in the natural science community since international standards have not been officially updated since 1953 (International Hydrographic Organisation, 1953). Examples are "Tropical Asian Seas" (Kleinherenbrink et al., 2017), "Southeastern Asian Seas" (Strassburg et al., 2015) and "Australasian Mediterranean Sea" (AMS; Tomczak and Godfrey 2003). We adapted the latter for the purpose of this study and refer to all marginal seas in the region, including Timor and Arafura Seas and also the SCS, as AMS. It will be useful to point to the SCS and the remaining part of the region, individually. We use the term "Eastern Archipelagic Seas" (EAS) for this purpose.

## 2 Methods

### 2.1 Ocean general circulation model experiments

We used a global ocean model configuration of the "Nucleus for European Modelling of the Ocean" (NEMO) code version 3.6 (Madec and NEMO-team, 2016). The model employs a global tri-polar ORCA grid and uses 46 vertical z-levels with varying layer thickness from 6 m at the surface to 250 m in the deepest levels. Bottom topography is represented by partial steps (Barnier et al., 2006) and is interpolated from 2-Minute and 1-Minute Gridded Global Relief Data ETOPO2v2[1] and ETOPO1[2](Amante and Eakins, 2009) for the coarse and high-resolution experiments, respectively. The atmospheric forcing for all experiments is

---

[1] http://www.ngdc.noaa.gov/mgg/global/relief/ETOPO2/ETOPO2v2-2006/ETOPO2v2g/, last access: 12 June 2018

[2] https://www.ncei.noaa.gov/access/metadata/landing-page/bin/iso?id=gov.noaa.ngdc.mgg.dem:316, last access: 11 October 2019

JRA55-do v1.3 (Tsujino et al., 2018) which builds on the JRA-55 reanalysis product but is specifically designed to be used as a forcing dataset for OGCMs. Most importantly, surface fields are corrected towards observations and heat and freshwater-fluxes are balanced with respect to a defined set of bulk formulas. Laplacian and bilaplacian operators were used to parameterize horizontal diffusion of tracer and momentum, respectively. The ocean model was coupled to the Louvain-La-Neuve sea-ice model version 2 (LIM2-VP; Fichefet and Maqueda 1997). To avoid spurious drifts of global freshwater content, all models used a sea surface salinity restoring with piston velocities between 137 mm per day and 40 mm per day. The coarse resolution experiments additionally used a freshwater budget correction that sets changes in the global budget to zero at each model time step.

In order to increase the horizontal resolution in the area of interest, a high-resolution "nest" was incorporated into the global base model. This approach allows a regional refinement of the horizontal resolution to $\frac{1}{20}°$ in the area from 50°S to 25°N and 75°E to 180°E. The nesting technique is based on the "adaptive grid refinement in fortran" (AGRIF; Debreu et al. 2008). A two-way communication was established between the high-resolution nest and the global "coarse" resolution model. The base model provided boundary conditions for the nest at every time step of the nest (the base model and the nest were integrated with a time step of 5 and 15 minutes, respectively). Due to the higher resolution of the nest in time and space, the boundary conditions were interpolated. The prognostic variables along the boundary of the nest were in turn updated after every time step of the base model by the calculations provided by the nest. Every three time steps, the full baroclinic state vector of the nest was averaged onto the base grid and fed back to the base model. This approach is well established, and the reader is referred for example to Schwarzkopf et al. (2019) for a detailed description of the procedure.

We integrated two reference configurations with interannually varying atmospheric forcing from 1958 to 2016:

- **REF005**: Reference run with $\frac{1}{4}°$ horizontal resolution globally and $\frac{1}{20}°$ in the nest area;

- **REF025**: Reference run with $\frac{1}{4}°$ horizontal resolution globally.

In addition, we followed the approach of Stewart et al. (2020) to construct a "quasi-climatological" forcing. More specifically, we extracted the recommended 12-month subset from May 1990 to April 1991 from the full forcing dataset and applied its individual fields repeatedly to suppress the interannual variability entirely or only for the computation of momentum or buoyancy fluxes, respectively. This way, we obtained three additional sensitivity experiments with $\frac{1}{4}°$ horizontal resolution globally, which were all integrated for 59 years to match the length of the reference configurations:

- **CLIM**: Full quasi-climatological forcing.

- **WIND**: Quasi-climatological buoyancy fluxes.

- **BUOY**: Quasi-climatological momentum fluxes.

The experiments WIND and BUOY are meant to isolate the momentum-flux-forced and buoyancy-flux-forced variability. The approach builds on the assumption that the total atmospherically-forced variability can approximately be understood by a linear combination of these two contributions. Because we only suppress variability on interannual and longer timescales, this assumption is not valid for variability on shorter timescales. A possible deviation could also arise from non-linearities and intrinsic, i.e. unforced, variability: an estimate of the magnitude of the latter is provided by the experiment CLIM. We note that the set of 1/4°-experiments (REF025, WIND, BUOY and CLIM) has been used previously for studies of Indian Ocean heat content (Ummenhofer et al., 2020), marine heatwaves (Ryan et al., 2021) and tropical Pacific sea-level variability (Wagner et al., 2021).

## 2.2 Observational datasets

We used two observational datasets to validate our model results. Satellite altimetry data is provided as a gridded product by the Copernicus Marine Environment Monitoring Service[3] (CMEMS) with a resolution of $\frac{1}{4}°$ that is available since 1993 (in contrast to our model simulations that start in 1958). Observations of temperature and salinity were taken from the World Ocean Atlas 18[4] (WOA18; Locarnini et al. 2019; Zweng et al. 2019) that is also available with $\frac{1}{4}°$ horizontal resolution and covers the period from 1958 to 2017.

## 2.3 Steric sea level

Sea-level changes are either due to changes of the total mass of the water column or due to density changes of seawater. The latter is referred to as steric sea level. Because seawater density is affected by temperature and salinity, steric sea level and can be separated into thermosteric and halosteric components, respectively. We diagnosed steric sea level changes from the stored model output:

$$\frac{\partial \eta^{steric}}{\partial t} = - \int_{-H}^{\eta} \frac{1}{\rho} \frac{\partial \rho}{\partial t} dz, \tag{1}$$

where ($\eta$) denotes sea level, $\rho$ is seawater density and $H$ is ocean depth. Thermosteric sea-level changes are given by:

$$\frac{\partial \eta^{thermosteric}}{\partial t} = \int_{-H}^{\eta} (\alpha \frac{\partial \Theta}{\partial t}) dz, \tag{2}$$

where $\alpha$ is the thermal expansion coefficient and $\Theta$ is potential temperature. In the same way, halosteric sea level variations can be expressed:

$$\frac{\partial \eta^{halosteric}}{\partial t} = \int_{-H}^{\eta} (\beta \frac{\partial S}{\partial t}) dz, \tag{3}$$

---

[3]https://resources.marine.copernicus.eu/?option=com_csw&view=details&product_id=SEALEVEL_GLO_PHY_L4_REP_OBSERVATIONS_008_047, last access: 12 June 2018

[4]https://www.ncei.noaa.gov/archive/accession/NCEI-WOA18, last access: 16 February 2021

where $\beta$ is the haline contraction coefficient and $S$ is salinity.

## 3   Results

First, we compare our model results to observational products for validation and to point out some resolution dependent biases. Sea level observations from satellite altimetry show strong interannual sea surface height[5] (SSH) variability in the western tropical Pacific, off the east coast of the Philippines islands. Within the AMS, variability is most pronounced in the deep basins of the SCS and the adjacent Sulu, Celeb and Banda Seas as well as in the coastal regions of the Arafura and Timor Seas (Fig. 1). Both hindcast simulations slightly underestimate the amplitude of standard deviations (SD) in particular in the western tropical Pacific by 1 to 2 cm (REF025 and REF005 respectively), but reproduce the spatial pattern well (Fig. 2 b, c). The spatial average of SSH over the AMS (see black contour in Fig. 2 c) reveals interannual to decadal variability. The latter is more pronounced in the high-resolution configuration (REF005) and agrees well with the observed positive trends since the early 90s (Fig. 2 d).

Figure 3 shows subsurface maps of mean fields and model biases of temperature and salinity with respect to observations from WOA18, averaged over the upper 400 m, from both hindcast simulations. Both hindcasts exhibit some temperature biases, in particular in the tropical Pacific, where opposite signs right on the Equator and at about 8°N (Fig. 3 a–c) indicate a large-scale southward displacement of the Intertropical Convergence Zone (ITCZ) in the Pacific that causes a southward shift of the equatorial upwelling regime. The coarse resolution hindcast REF025 also shows positive temperature and salinity biases in the AMS. In particular in the northern part of the SCS, where values exceed 3°and 0.3 psu (Fig. 3 b, e). These biases are greatly reduced in the high-resolution setup to temperature biases well below 0.5°C and salinity biases below 0.1 psu.

### 3.1   Impact of buoyancy fluxes on SSH

We now turn to the forcing mechanism of decadal SSH variability and inspect the low-pass filtered (cutoff period of 8 years) SSH timeseries. Figure 4 presents maps of standard deviation (SD) for all five experiments, as well as SSH anomalies averaged over the AMS. The two hindcast simulations show a close resemblance in terms of amplitude and pattern, suggesting only a minor influence of the model resolution (Fig. 4 a, b). The pattern and amplitude of the WIND experiment (Fig. 4 c) closely match the signal in REF025, indicating that wind stress variability is the most important driver of the variability. The SSH signal driven by the interannual variability of buoyancy fluxes shows much lower amplitudes and a different spatial pattern (Fig. 4 d; note the different colorbars): it is most pronounced outside the AMS in the Kuroshio region and the southeastern, tropical Indian Ocean and within the AMS in the SCS, whereas values in the EAS are relatively low (about 0.3–0.4 cm).

While in most areas the total variability in REF025 can be understood as a linear superposition of variability due to momentum and buoyancy fluxes, this is not the case in the Kuroshio region or the Gulf of Thailand. Possible reasons are cancelling

---

[5]We will refer to the sea-level estimate of the OGCMs, which gives height above the geoid, as SSH

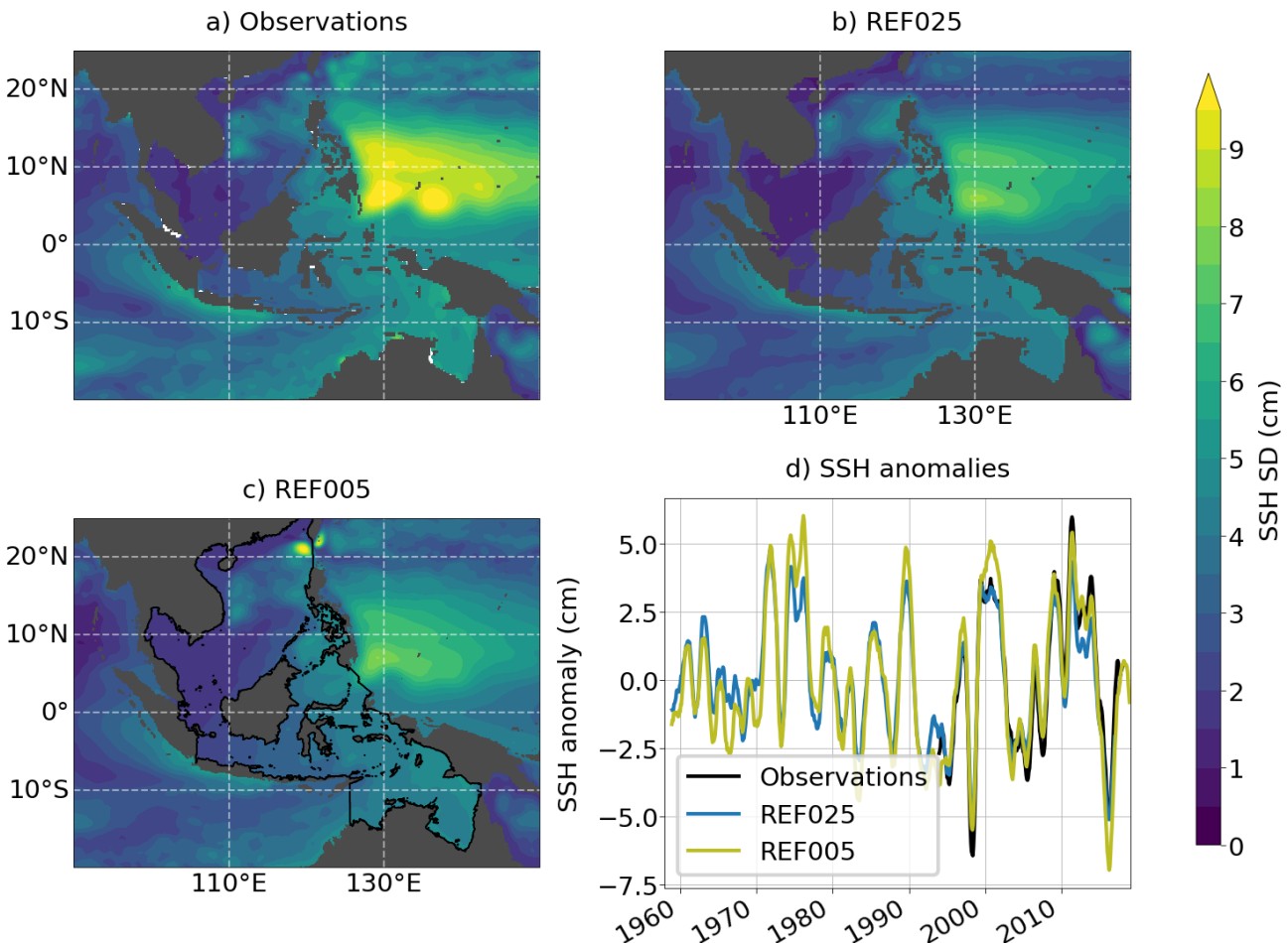

**Figure 2.** *SD of SSH from (a) observations and two hindcast experiments (b) REF025 and (c) REF005 over the period 1993 to 2016. d) SSH anomalies averaged over the region marked by black contours in panel c). The global mean trends over the integration period* have been subtracted, and all data were smoothed with a 12-month running mean window.

effects of out-of-phase variations of momentum and buoyancy fluxes or, as pointed out above, intrinsic ocean variability and non-linear effects. As evident from Fig. 4e), intrinsic variability in the AMS is below 0.2 cm everywhere except in the SCS.

Amplitudes in the northern SCS reach values of up to 1 cm and are thereby comparable to the values of the buoyancy forcing experiment and account for about 25 % of the total variability in REF025.

Averaged over the AMS, buoyancy fluxes amplify the wind-stress-driven variability (Fig. 4 f). If we consider the change in variability between REF025 and WIND, rather than the variability in BUOY itself, to be the effect of buoyancy fluxes,

we find a contribution of 9% (SD of 1.54 cm in REF025 and 1.4 cm in WIND) by which buoyancy fluxes amplify the wind

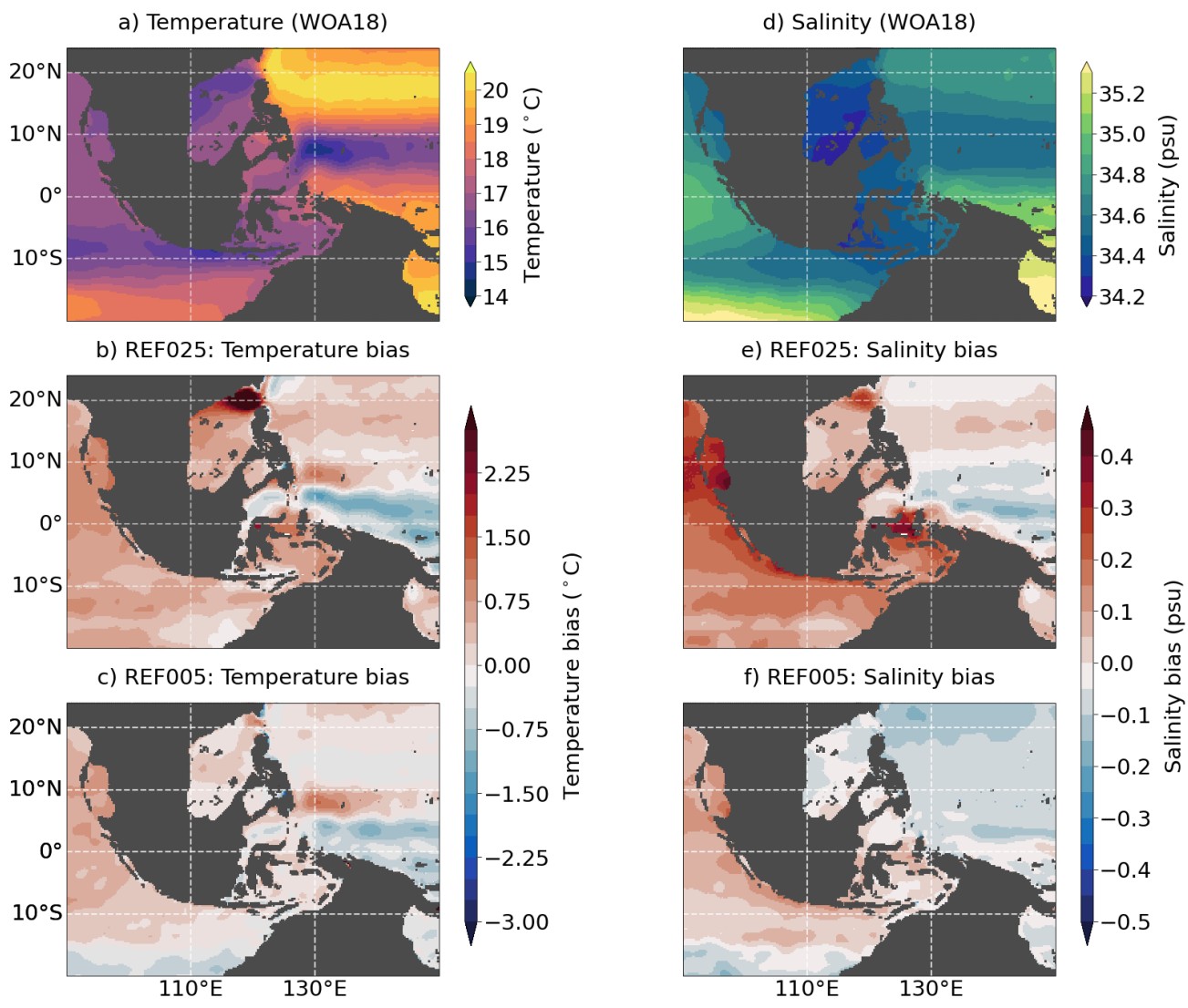

**Figure 3.** *Mean fields (1955–2017; upper row) and model biases (1958–2016; middle and bottom row) averaged over the upper 400 m for (a-c) temperature and (d-f) salinity. Note that data is only shown where the water depth exceeds 400 m.*

stress-driven-variability. This fraction is however variable over time and reaches values of up to 50 %, for example during the late 1990s. When estimating the effect of buoyancy fluxes from BUOY directly, we obtain a much larger contribution of 24 %. The reason for this is intrinsic variability that is included in all experiments and adds to the variability in BUOY. We found no indication that buoyancy fluxes contribute to the increase in sea level between roughly 1990 and 2010 that is visible in both

hindcast simulation and WIND.

## 3.2    Linear regression of SSH on ENSO and PDO indices

A linear regression model is used to quantify the impact of low-frequency ENSO changes and PDO cycles on sea-level variabil­ity. We use a band-pass filtered Niño3.4 index[6] (8–13 years) and a low-pass filtered PDO[7] index (13 years) as predictors. Both

indices are derived from REF025 and compare well with observational estimates (Fig. A1). Both indices are not significantly correlated (at a confidence interval of 5%) by construction because their variability is limited to time scales longer or shorter than 13 years. This allows a linear superposition of the responses and their respective coefficients of determination ($R^2$ gives the fraction of variance explained by a linear model; e.g. Thomson and Emery 2014). As before, SSH data has been low-pass filtered (8 years) prior to the regression. Note that the PDO can be considered the low-frequency component of ENSO in this

context to the extent that the PDO can be regarded as the low-frequency component of ENSO, an alternative, and largely equiv­alent separation could be build on the low-pass filtered Niño3.4 index instead of the PDO index. We will get back to this point in section 4. The Indian Ocean Dipole Index (IOD; Saji et al. 1999) could account for possible impacts of the Indian Ocean. We do not include the IOD in our analysis because the index shows only very weak variability on the timescales considered here (not shown).


Figure 5 depicts the linear responses of SSH from three experiments to the Niño3.4 and PDO indices. Since its results are very similar to REF025, we omit the WIND case here but defer it to the appendix (Fig. A2). REF025 and REF005 show a sim­ilar response to positive ENSO cycles (Fig. 5a, b). Strong negative anomalies in the western tropical Pacific with amplitudes of 4 cm off the east coast of the Philippines islands leak into the north-eastern part of the SCS and also follow the equatorial wave

guide across the Celebes and Banda seas and along the Australian shelf break into the Indian Ocean. Here, both hindcasts find amplitudes between 1 and 2 cm (Fig. 5 a, b). This pattern is mostly determined by wind stress variability, as the wind stress experiment shows similar results (Fig. A2) and the pattern in BUOY differs strongly (Fig. 5 c). Here, SSH shows a weak but non-negligible, negative response to ENSO cycles which is uniform across the domain and thereby amplifies the wind-stress-driven response by about 0.2–0.3 cm in most parts of the region. Buoyancy fluxes counteract the wind-stress-driven variability

off the northeast coast of Australia, to the south of Papua New Guinea, such that the effective ENSO related SSH variability is

---

[6]The Niño3.4 index is defined as the area averaged sea surface temperature (SST) anomaly in the Niño3.4 region (5° N–5° S, 170° W-120° W) with respect to the monthly climatology and normalized by its SD

[7]The PDO index is obtained via an Empirical Orthogonal Function of monthly SST anomalies in the Pacific north of 20° N and defined as the leading mode of variability. The climatological annual cycle is subtracted to remove long-term trends.

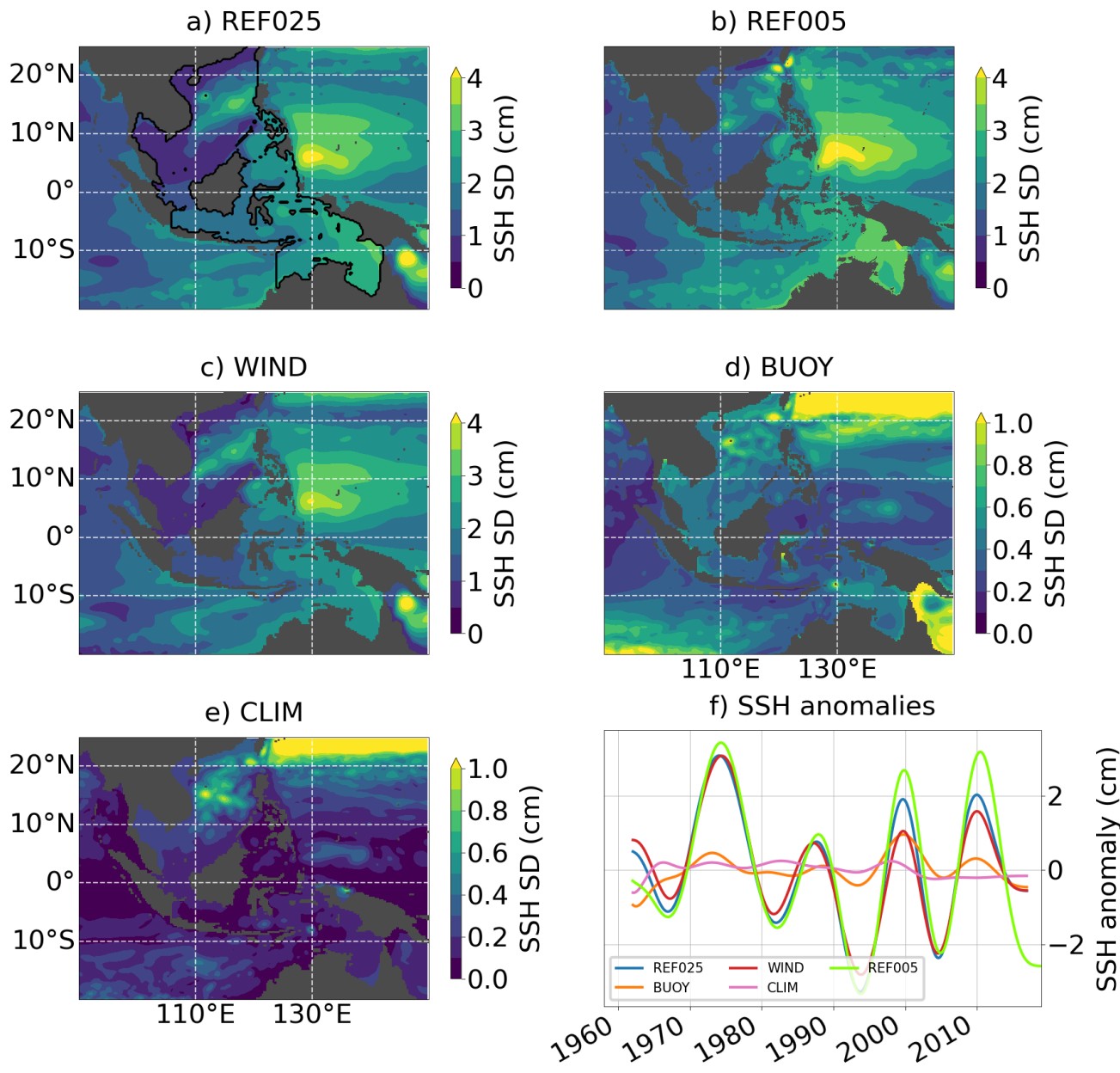

**Figure 4.** *SD of low-pass filtered SSH timeseries (8 years) from (a) REF025, (b) REF005, (c) WIND, (d) BUOY, (e) CLIM and (f) low-pass filtered SSH anomalies averaged over black contour lines shown in panel a). Note the different colorbars.*

almost zero.

The linear response to the PDO index shows strong amplitudes of 3 and 4 cm (in REF005 and REF025 respectively; Fig. 5 d, e) to the east of Papua New Guinea in the region of the South Pacific Convergence Zone (SPCZ). Anomalies stretch northwestward along the shelf and into the EAS, in particular into the Arafura seas, with values between 1 and 2 cm. Both hindcast experiments also suggest a PDO-imprint in the Java seas and the central SCS. Again, the variability is mostly driven by momentum-fluxes (Fig. A2). Buoyancy fluxes can not explain the dominant pattern and tend to dampen the variability regionally. In particular in the SPCZ, but also in the western tropical Pacific north of 10°N, in the northern part of the SCS and in the Sulu Sea. The response in the remaining regions is mostly below 0.1 cm.

In order to determine what fraction of the total variability can be explained by these two climate indices, Fig. 6 displays $R^2$. Both hindcasts (and also the WIND experiment; Fig. A2 a) agree that ENSO determines a large fraction of the low-frequency variability in the western tropical Pacific at around 5°N and in all deep basins with values of over 80 % and, to a slightly lesser extent, also in the Timor Sea and along the Australian shelf break. This fraction is reduced to less than 50 % for the buoyancy flux related variability (Fig. 6 c).

Both hindcast simulations find that over 50 % of the total decadal SSH variability in the SPCZ and the Java Sea and over 30 % in the central SCS can be explained by PDO cycles. Both differ in their estimate of the quantitative importance of the PDO for the remaining region of the EAS, but still find 10–20 % of explained variance in case of REF005 and even more in REF025. Note, however, that REF025 showed temperature and salinity biases in this region that might be the reasons for these discrepancies. The PDO does not drive any buoyancy-flux-driven SSH variability in the AMS (Fig. 6 f). The linear combination of both indices (not shown) gives high scores of over 80% throughout the domain except in the central and southern part of the SCS where intrinsic variability is relevant.

## 3.3 Decomposition of SSH variability into thermosteric and halosteric contributions

To further elucidate the mechanisms shaping the sea-level signal, it is useful to analyse the dynamical response and also decompose the sea-level signal into its steric components. We neglect sea-level anomalies due to mass fluctuations because their relative contributions are small everywhere except on shelf regions (not shown; e.g. Forget and Ponte 2015) where the total variability is weakest.

As expected, the dynamical response to the ENSO cycles (Fig. 7 a-c) is governed by anomalies of the North Equatorial Counter Current at 4°N and by thermosteric SSH anomalies of over 4 cm in the western tropical Pacific. The latter change the inter-basin pressure gradient between the Pacific and the Indian Ocean. Specifically, a positive ENSO cycle leads to a reduction in SSH, which causes a decreased pressure gradient across the AMS. Consequently, northward velocity anomalies

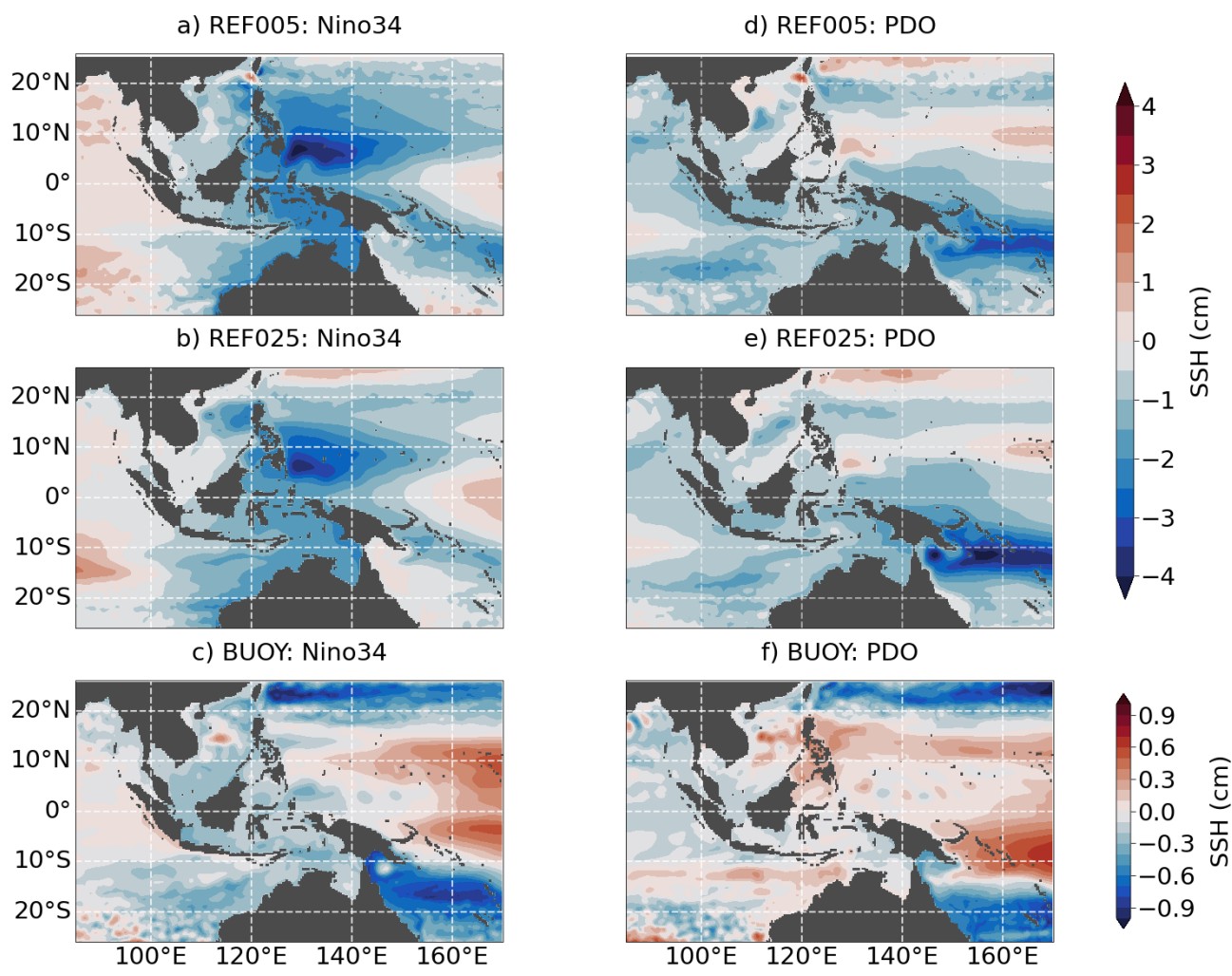

**Figure 5.** *Linear regression of SSH with (a-c) Niño3.4 index and (d-f) PDO index for two hindcast (REF025, REF005) and buoyancy forcing experiments (BUOY). Climate indices are derived from its base model in case of REF005 and from REF025 for the other two experiments. All data has been filtered with an 8-year low-pass filter. Note the different colorbars.*

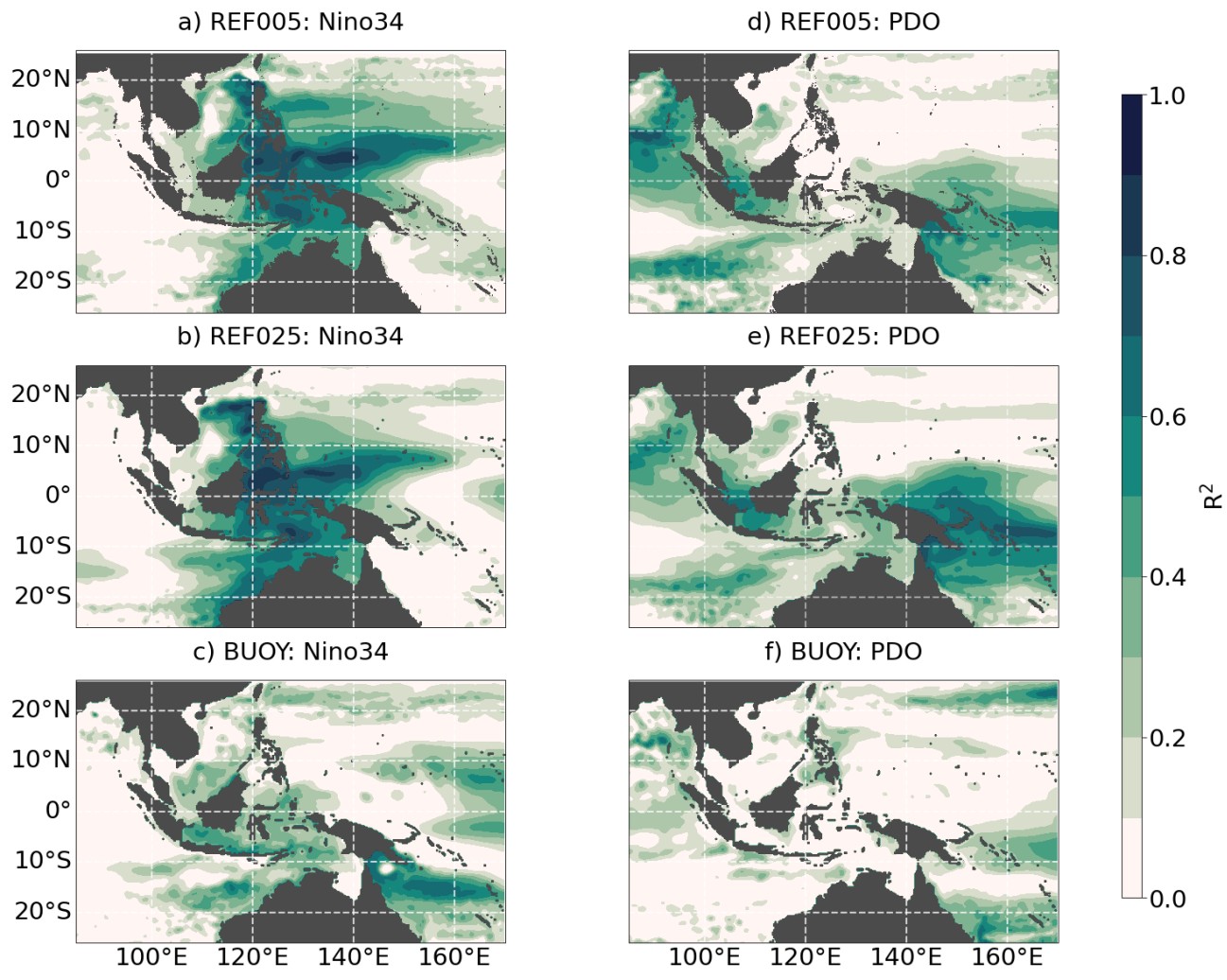

**Figure 6.** *Coefficient of determination for the linear regression of SSH with (a-c) Niño3.4 index and (d-f) PDO index for two hindcast (REF025, REF005) and buoyancy forcing experiments (BUOY). Climate indices are derived from its base model in case of REF005 and from REF025 for the other two experiments. All data has been filtered with an 8-year low-pass filter.*

in the Makassar Strait and Banda Sea indicate a reduction of ITF transport. Steric changes are consistent with this dynamical response of the upper ocean. The reduction of the ITF transport causes a reduction in heat and freshwater transport from the western tropical Pacific through the EAS, which causes negative thermosteric and halosteric SSH anomalies along the ITF pathway. In particular in the Banda Sea and Timor Sea, halosteric anomalies complement the thermosteric signal and make a relevant contribution ($\sim$1 cm) to the total signal of over 2 cm. Note, however, that thermosteric anomalies seem to originate

from the warm pool region in the western tropical Pacific whereas halosteric anomalies are most pronounced in the EAS and the ITF outflow region in the Indian Ocean. This aspect is elaborated further two paragraphs below. Another question concerns the possibility of an asymmetric response to positive and negative ENSO phases. From an analysis of their individual contributions, we find that almost all regions with a non-negligible response to ENSO are characterized by similar amplitudes of SSH variability during both phases. Only the Arafura Sea responds stronger to the negative cycle (not shown).


PDO related changes are dominated by anomalies of the South Equatorial Current and the well-known SSH changes in the SPCZ region (Fig. 7 d-f). Negative SSH anomalies during a positive PDO cycle are due to thermosteric anomalies, which are partly compensated by halosteric contributions. Negative anomalies change the pressure gradient along the coast of New Guinea and weaken the equatorward transport of warm and saline South Pacific waters via the New Guinea Coastal Current.

This produces unfavourable conditions for South Pacific waters to enter the EAS, where we find negative thermosteric anomalies that are partly compensated by positive halosteric anomalies, similar to the changes in the SPCZ.

The compensating effect of temperature and salinity changes during PDO cycles also manifests in the vertical profiles of temperature and salinity in the AMS. Figure 8 shows linear regressions of T and S averaged over the Banda Sea (7°S-3°S,

125°E 133°E: see red box in Fig. 7). Both hindcast-experiments (Fig. 4a, b) indicate a strong, negative subsurface response (hence the density compensation) to positive PDO cycles. The ENSO response is dominated by a subsurface temperature anomaly causing negative SSH values. The subsurface temperature anomalies during both ENSO and PDO cycles are amplified by sea surface salinity (SSS) anomalies. The two sensitivity experiments allow some insight into the origin of these surface signals (Fig. 4a, b). The momentum flux experiments (WIND) reproduces the subsurface signals, highlighting the fact that they

are due to wind-stress-driven advection, but shows no surface anomalies associated with ENSO and PDO cycles. These are instead driven by buoyancy fluxes. This fits the ENSO-related halosteric anomalies (Fig. 7c) that do not seem to originate in the western tropical Pacific, but are instead forced by local buoyancy fluxes.

## 4 Summary and discussion

### 4.1 Summary

We determine the characteristics of decadal sea-level variability in the Australasian Mediterranean Sea and their relation to large scale climate modes of ENSO and PDO by using a series of global ocean model experiments. Two reference configura-

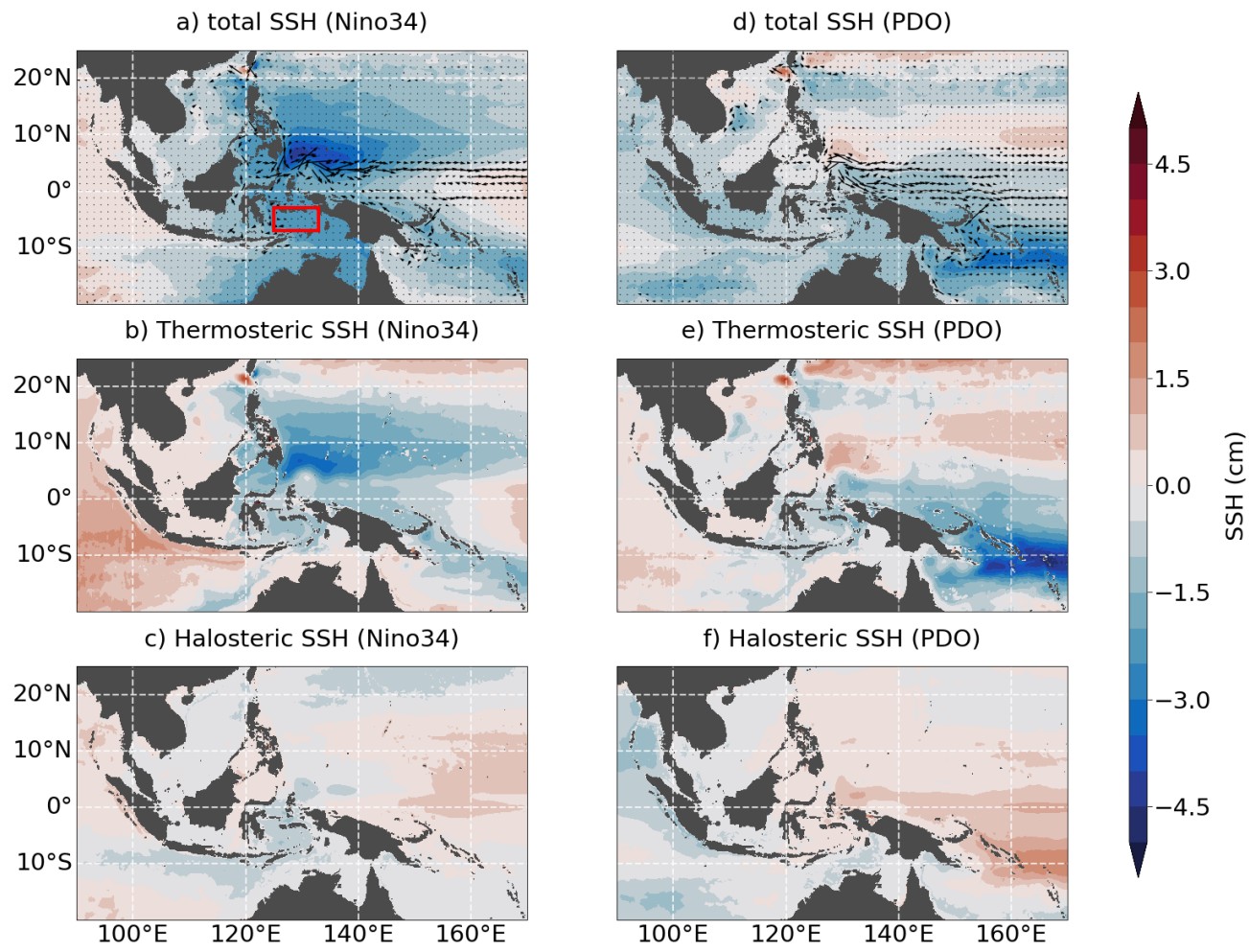

**Figure 7.** *Linear regression with (a-c) Niño3.4 and (d-f) PDO index of upper ocean currents (arrows; 0–150 m), total, thermosteric and halosteric SSH. All data is taken from REF005 and were filtered with an 8-year low-pass filter. Note that the colour shading in panels a) and b) is the same as in Fig. 5a, b)*

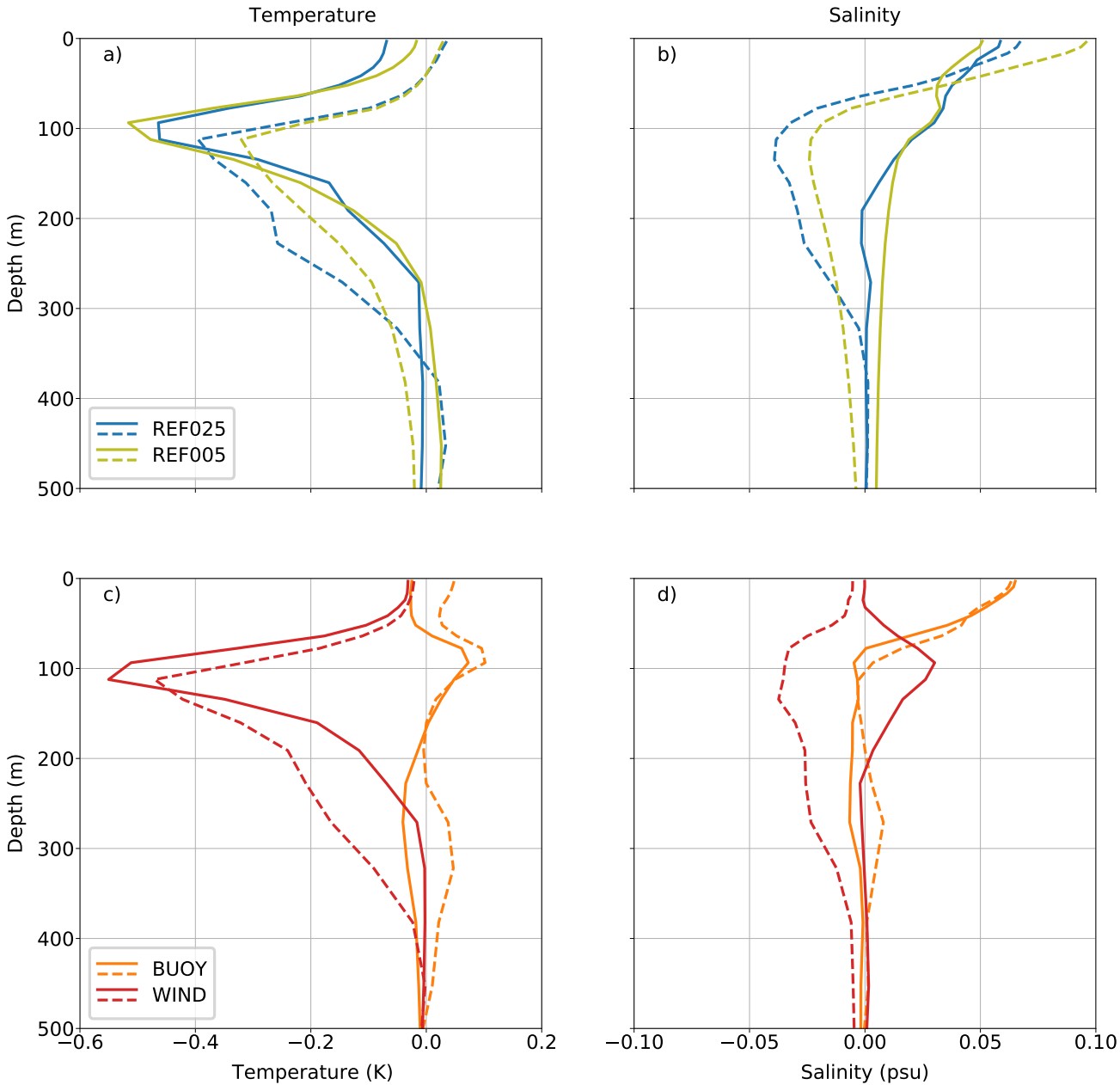

**Figure 8.** *Linear regression of temperature and salinity to Niño3.4 (solid) and PDO index (dashed) from (a, b) reference experiments and (c, d) sensitivity experiments. All data has been filtered with an 8-year low-pass filter.*

tions, that resolve the area of interest with $\frac{1}{4}°$ and $\frac{1}{20}°$ horizontal resolution, are set up and forced with the newly developed JRA-55 do forcing (Tsujino et al., 2018). Both simulations are able to reproduce observed sea-level variability in the western tropical Pacific and the AMS. We find minor resolution dependent differences between both simulation in terms of SSH. However, the coarse resolution configuration suffers from some upper ocean temperature and salinity biases, in particular in the northern part of the SCS and the Celebs Sea, which are alleviated in the high-resolution experiment.

We assess the regional fingerprints of ENSO and PDO on SSH in the region. Both climate modes produce strong sea-level signals in the western tropical Pacific. The ENSO-related signal is most pronounced just north of the Equator south of $10°N$, which is in line with previous studies (e.g. Meng et al. 2019; Han et al. 2019). These strong SSH signals cause geostrophic flow anomalies as they change the inter-basin pressure gradient between the Pacific and the Indian Ocean and weaken the ITF (Hu et al., 2015). The imprint of the PDO in the northern part of the warm pool is weak and instead shifted southward. This confirms earlier reports which found the effect of the PDO (on multidecadal time scales) in the western tropical Pacific to be most pronounced in the region of the SPCZ (Han et al., 2019; Moon et al., 2013; Becker et al., 2012). Our results suggest a partial compensation of thermo- and halosteric PDO-related SSH anomalies. This is also in accordance with previous studies, which report a cooling and freshening of the southern part of the warm pool in response to a positive PDO cycle (Cravatte et al., 2009).

Within the AMS itself we find that, except for the central and northern SCS, over 80% of the variability in the deep basins of the AMS is determined by low-frequency fluctuations of ENSO. The imprint of PDO cycles within the AMS is pronounced in the shallow Arafura/Timor and Java seas, where it explains 20% and 50% of the total variability, respectively. Analysing sea-level trends since the early 90s from altimetry, tide gauge observations and reconstructions, Strassburg et al. (2015) find similar results with respect to the PDO. Furthermore, our results suggest distinctly different patterns of variability related to ENSO and PDO in the SCS, where the ENSO pattern is limited to the northeastern part, off the west coast of Luzon, and the PDO causes anomalies in the central SCS. This confirms previous results of idealized model studies (Cheng et al., 2016).

It can be expected that a clarification of the spatial pattern of multidecadal variability will aid future projections of sea level in the region. With the recent phase shift of the PDO, the affected regions should be primed for a weakening of the recent sea-level rise, as it is also observed in the western tropical Pacific (e.g. Hamlington et al. 2016; Piecuch et al. 2019).

In addition to two hindcast simulations, we set up a series of sensitivity experiments that used quasi-climatological forcing (Stewart et al., 2020) to remove interannual variability from the full forcing or only from the momentum or buoyancy fluxes. To our knowledge, the impact of buoyancy fluxes and intrinsic variability on regional sea level in the region has not been described in previous studies. Our results suggest that, even though momentum fluxes are the primary driver of variability, buoyancy fluxes are not negligible. According to our findings, they only contribute around 10% on average to the total variability. However, this fraction is highly variable and can increase to up to 50% during ENSO cycles when they amplify the wind-stress-driven sea-level anomalies. Indeed, buoyancy fluxes have been shown to be important for sea-level variability in

other regions, including the tropical Pacific (Piecuch and Ponte, 2012; Ponte, 2012; Wagner et al., 2021). Intrinsic variability is relevant in the central SCS, where it accounts for 25% of the low-frequency variability.

## 4.2 Discussion

We did not investigate the origin of the high levels of intrinsic variability in the SCS in detail and can only speculate at this point. A connection to mesoscale variability and the eddy-active Kuroshio region, that also shows high levels of intrinsic variability (Fig. 4 e), seems likely. Both regions are connected via the Luzon strait between Taiwan and the Philippines, and both regions show high levels of eddy kinetic energy (EKE; e.g. Scharffenberg and Stammer 2010). Besides local (Wang et al., 2008) and remote (Chen et al., 2009; Cheng and Qi, 2010) wind forcing, the Luzon strait transport (LST) has been identified in previous studies to drive interannual EKE variability in the SCS. An increased LST leads to a strengthening of the mean flow in the SCS, which creates favourable condition for baroclinic instabilities that allow a downscale energy transfer (Sun et al., 2016). The strength of the LST is closely related to the Kuroshio current intrusion into the SCS (Qu et al., 2004; Wang and Fiedler, 2006) and the strength of the Kuroshio current itself. The Kuroshio current flows poleward along the western boundary, to which the Luzon Strait poses a gap through which the current can enter the SCS. The mechanism that control the pathways and strength of this intrusion are complex and remain controversial (Nan et al. 2015 and references therein), but also include mechanism by which eddies, generated by intrinsic variability in the Kuroshio region, impact the LST. While eddies are unlikely to cross the Luzon Strait directly, they have been observed to modulate the strength of the Kuroshio current (Jie and De-Hai, 2010), which could be a means by which intrinsic variability from the Kuroshio region is indirectly communicated into the SCS. However, the viability of this hypothesis and the determination of the mechanism involved need to be addressed in future research.

The aim of this study is to identify deterministic low-frequency pattern of variability, associated with large-scale climate modes. Although the separation of climate indices along time scales (ENSO: 8–13 years; PDO: > 13 years) is somewhat arbitrary, the results are not overly sensitive to the choice of the separation frequency. In fact, using only a low-pass filtered ENSO index (> 8 years) yields results that are similar to the combined effect of bandpass filtered ENSO and low-pass filtered PDO we presented here. As already mentioned above, the PDO can be understood as the low-frequency component of ENSO in this context, and separating the time scales allowed us to show that the spatial pattern of SSH variability in the AMS, and the mechanisms that drive it, differ over these time scales.

An uncertainty of this study is the atmospheric forcing product used to drive the OGCMs. McGregor et al. (2012) demonstrated that most available atmospheric reanalyses, in conjunction with a linear shallow water model, are able to reproduce the large-scale pattern of SSH variability in the Pacific, and we can confirm this for the dataset used here. However, we find minor biases in both reference configurations that are most likely due to the forcing. Hsu et al. (2021) report a wind stress curl bias in JRA55-do in the Pacific between 4–9° N that acts to diminish the meridional SSH gradient and flatten the SSH trough at this

latitude. This is consistent with the warm bias we found at this latitudinal band. Also, consistent with the results found here, they further report an underestimation of SSH trends between 1993 and 2007 in the western tropical Pacific and relate it to an underestimation of the observed trade wind intensification.


A final point concerns the impact of model resolution in studies of variability patterns in this region. Although in our study we found only minor resolution-depended differences between the $\frac{1}{4}°$ and $\frac{1}{20}°$-configurations with respect to SSH signals, the stronger biases in the temperature and salinity distributions seen in the coarser simulation should be taken as a reminder that a realistic representation of the dynamics of the AMS area with its complex, small-scale bathymetry can require a very high

model resolution. The importance of resolving the narrow straits and bathymetric details for capturing the ocean dynamics in the AMS was recently stressed in the review by Xue et al. (2020) of coupled ocean-atmosphere modelling for this region. In this context, it should be noted that the "coarse" 1/4°-grid used in this study would rather be in the higher-resolution category of current climate simulations with coupled ocean-atmosphere models, suggesting that projections of regional trend patterns in this area need to be interpreted with due caution. Downscaled projections (e.g., Sun et al., 2012; Feng et al., 2017), in which

an uncoupled high-resolution OGCM is forced with a long-term climate change signal obtained from a climate model, might be a way to address this issue.

*Data availability.* Data shown in this paper are available at https://data.geomar.de/downloads/20.500.12085/e300e837-c02e-4939-a9b6-a9be163cbd26/

# Appendix A: Appendix

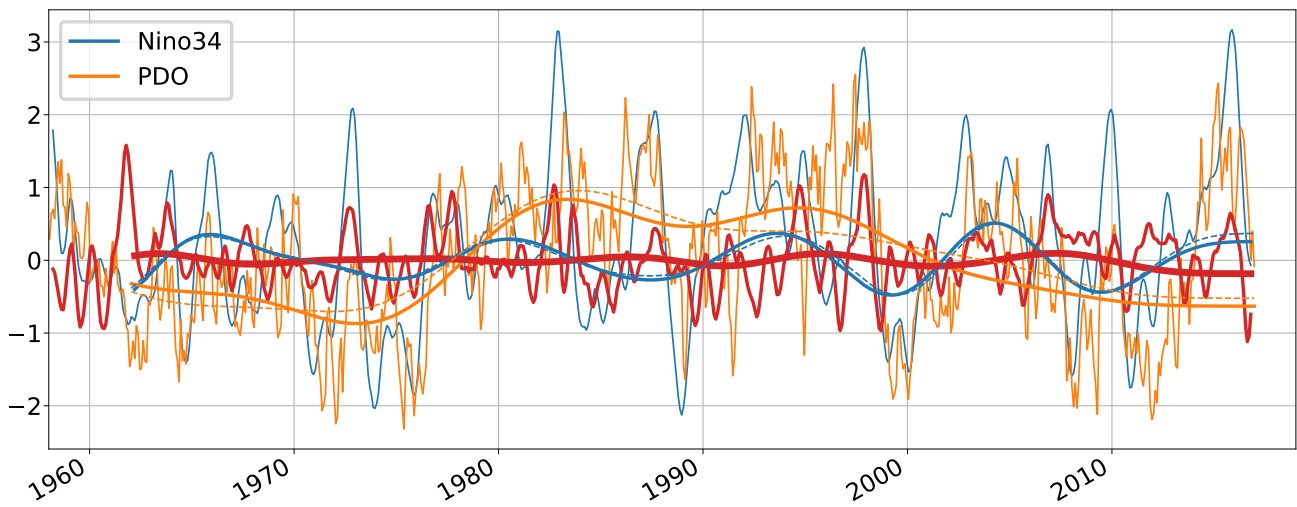

**Figure A1.** *Niño3.4 index and PDO index, both calculated from REF025. Smoothed lines are band pass filtered with cutoff periods of 8 and 13 years (Niño3.4 ) and low-pass filtered with cutoff periods of 13 years (PDO). Dashed lines show observational estimates of the PDO index (http://research.jisao.washington.edu/pdo/PDO.latest, last access: 08 Jun 2017; Mantua et al. 1997) and the Niño3.4 index (https://psl.noaa.gov/gcos_wgsp/Timeseries/Data/nino34.long.data, last access: 07 Jun 2017; Rayner et al. 2003) )*

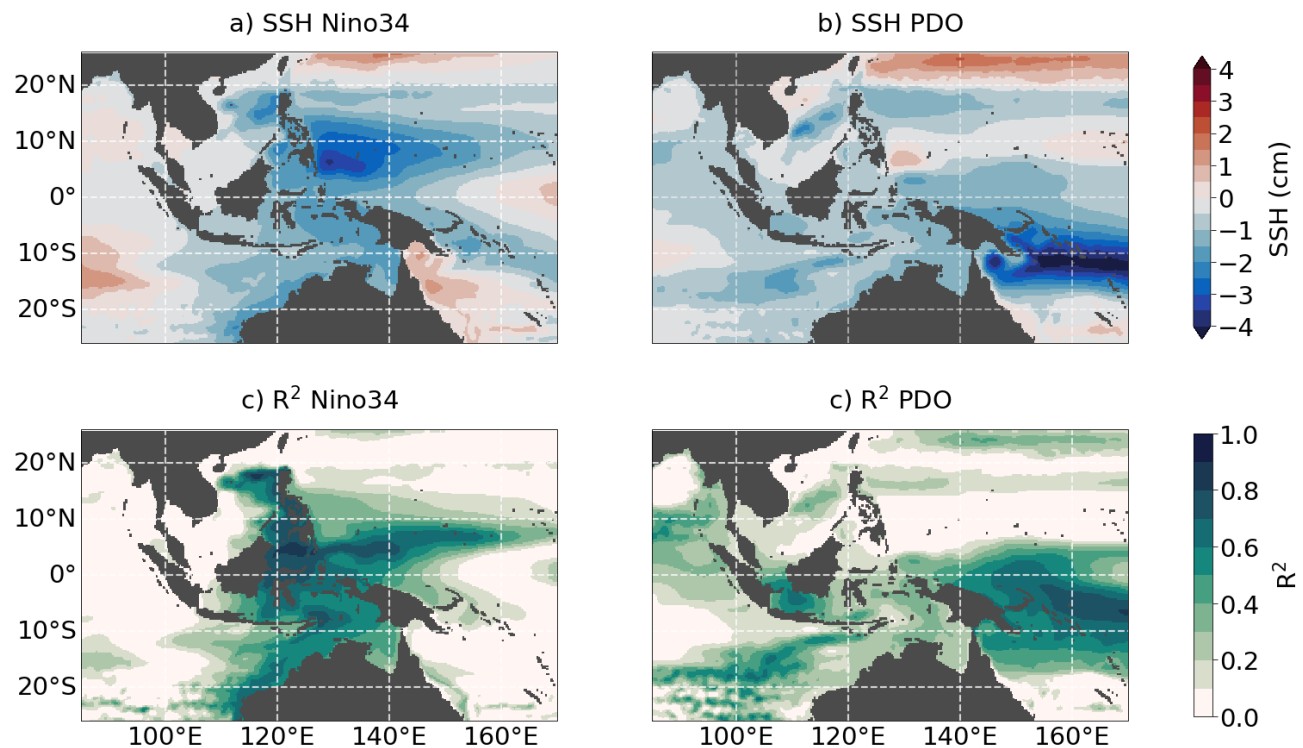

**Figure A2.** *Linear regression of SSH from WIND on (a) Niño3.4 and (b) PDO index and (c, d) respective coefficients of determination. Both climate indices are derived from REF025. All data were filtered with an 8-year low-pass filter.*

*Author contributions.* PW and CB defined the overall research problem and methodology; PW developed, ran, and validated the OGCM experiments ; PW produced all figures; PW prepared the paper with contributions from CB.

*Competing interests.* The authors declare that they have no conflict of interest.

*Acknowledgements.* PW was supported by the Deutsche Forschungsgemeinschaft (DFG) as part of the Special Priority Program (SPP)-1889 "Regional Sea Level Change and Society" (Grant BO907/5-1). All model simulations have been performed at the North-German Supercomputing Alliance (HLRN). We thank Markus Scheinert for his contribution to the ocean model simulations used in this study. We thank Brett Buzzanga and two anonymous referees for their constructive comments.

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
