# Peer review of "Decadal sea-level variability in the Australasian Mediterranean Sea"

_Ocean Science, 2021_

## Referee Comment (RC3)

Review – Decadal sea level variability in the Australasian Mediterranean Sea
Specific (minor) comments:

1. ENSO and PDO are not defined (neither in the abstract or the main text). Although they are well-known acronyms, it should be defined.
2. Whenever talking about sea-level trends, it is important to mention the period that it referred to. For example, L24 a trend is given, but no period. Also the first sentence of the paper is 'over the last three decades', but then the older reference is 2013, so I assume you mean 1980-2010? Not including 2010-2020? Adding the period to the sentence would clarify this.
3. L22 you mention that some areas in the western tropical pacific have received a lot of attention. Maybe adding a more specific example would be nice.
4. L34: You mention the 'Luzon Strait'. I have no idea where it is. Having a study are map, with such names would help.
5. L35 'might amplify the signal'. Which signal?
6. L43-44: is the ITF variability governed large-scale climate events (like ENSO and PDO), or by the climate indices (like Niño3.4)?
7. L52: Wijffels and Meyers showed (instead of 'could show'), based on XBT observations, that sea-level anomalies… Would also be good to mention which period of observations they used.
8. L57: You mention 'mass and steric components'. I think it would be good to define to the reader what exactly are these components (not everyone will know). And mainly the steric contribution could be better explained, as you discuss it further on your results.
9. L61: are you talking about sea-level trends? ('reconstructed decadal trends during the…'). Make it clear to the reader.
10. The term halosteric appears for the first time on Line 67, and then again with the term thermosteric in the objectives of the paper (Line 83). But you didn't introduce what thermo and halosteric changes are. Although the readers of Ocean Sciences might be familiar with it, in my opinion you could briefly define thermo and halosteric changes before having it as one of the objectives of the paper.
11. Line 75 should be connected to the previous paragraph.
12. Line 104 should be connected to the previous paragraph.
13. Lines 104-106: Won't this freshwater budget correction affect then your halosteric analysis?
14. L122: is there a specific reason for using the years 1990-1991 as your climatological forcing?
15. L129: Add 'respectively' at the end of the sentence: 'isolate the momentum forced and buoyancy forced variability, respectively'.
16. L134 should be connected to the previous paragraph.
17. L139: Here you start about your validation with observations. It's important to highlight that you are using satellite altimetry observations, and that they only start in 1993
18. L143: Here and in the rest of the paragraph I think you meant figure 1?
19. L149: biases in relation to what? Model biases? Make it clearer ☺
20. L151: What is the reference period of the mean fields? Is it the same as the model biases (which I am assuming is for the entire period?)

21. Figure 2: Here locations where you don't have data (which is a lot of the AMS actually) has been plotted with the same color as land, which was confusing. So I suggest you to plot it in a different color. I was also left wondering why you don't have data there (I can understand from the observations, if are areas that are too shallow… but I would expect the model to have values there as well). So it might be worth adding one sentence about it in the main text and/or in the caption of the figure.

22. L162: I found the word 'confirming' a bit of a bold statement, I suggest replacing it with 'indicating'.

23. L164 and Figure 3: Here you sat that you are using different colorbar for the same figure. I can understand why you want to do this, since if you keep the range of panel d and e to 0-4 we won't be able to see the patterns… But I still find it a bit misleading. Even though the colorbars are indicating that they range only up to 1, I find it harder to compare the panels, which is a bit the point of the plot. So it's up to you, but I would suggest fixing the colorbars throughout the figure.

24. Both paragraphs at L167-179 are just talking about REF025 experiment. What about REF005?

25. L185: 'e.g.' should be followed by a comma. And the first name of the authors of the reference are appearing (Thomson, R.E.)

26. L190; Bring the reference to Figure 4 here ('REF020 and REF005 show a similar response to positive ENSO cycles (Figure 4.a,b)'.

27. Figure 3: Why are the grid lines here marking only 10˚N, 0˚ and 10˚S? And the longitudes are on a different spacing than in Figure 1 and 2. This is very small detail, but you could use always the same gridline spacing.

28. L200: Bring the reference to Figure 4d.e. here: 'the linear response to the PDO index (Figure 4.d.e.)…'

29. L200: I think you meant a 'and' here instead of 'to': 'strong amplitudes of 3 and 4 cm'.

30. L212: Figure 5d is about PDO… I thought you were talking about ENSO, so figure 5c?

31. L2017: 'PDO does not drive any buoyancy flux driven variability', in the AMS right? Because I can seem some variability on the surrounding seas around the AMS.

32. L224: How do you know that the mass fluctuations are small? Is this based on previous research, if so, then reference it.

33. Figure 5: Add variable name to the colorbar (R-squared), and also mention in the caption which coefficient you are showing (R-squared)

34. L233: halosteric anomalies complement (not amplify) the thermosteric signal (it is not because of the halosteric variation that the thermosteric signal will be higher, but you will have a total steric change that is higher).

35. L245: Remove 'does': changes during PDO cycles also manifest in the vertical profiles'.

36. Figure 6 caption: linear regression with (instead of to); move the 'of upper ocean currents' to the end of the sentence or to another sentence (right now it seems you made the correlation with the currents, but I believe you are just showing the currents). And are those the mean current velocity? Make it clearer.

37. First paragraph of 'Summary and conclusion' should be connected to the second one.

38. L269: pressure gradient between the Pacific and (not in) the Indian Ocean.

39. L270: Remove 'again'.
40. Line 289 should be connected to previous paragraph.
41. L306: Add 'current' with Kuroshio here and in the following mentions of the current (Kuroshio current flows…)
42. L324: 'McGregor demonstrated (not could demonstrate).
43. General conclusion statement: The resolution effect is clear for the biases, but it doesn't seem to have had a significant impact in your SSH variability analysis. I'm not sure if your final and general conclusion should be about this.

---

## Author Response (AR1)

We would like to sincerely thank Brett Buzzanga for the time and effort he put into this review and the helpful suggestions improving our manuscript. We address the raised issues below.

**General Comments**

**Overall, "Decadal sea level variability in the Australasian Mediterranean Sea" is very well constructed, explained, and is an important contribution to our understanding of how climate variability impacts regional sea level. Specifically, the authors use an ocean model to determine how ENSO and PDO impact decadal regional sea-level variability in the Australasian Mediterranean Sea (AMS), determine the contributions of momentum and buoyancy fluxes to sea-surface height (SSH) variability, and further disentangle the steric variability into thermo and saline components. I do not see any major flaws in methodology or reasoning, and as such, recommend acceptance after addressing some minor issues mainly related to presentation.**

**Specific Comments**

1. **It would be very helpful to include an overview map of the study area, with the different seas and currents discussed clearly marked. The labels in Fig. 1a were helpful but not sufficient in this regard.**

We added a map of the region as a new Fig. 1.

[Figure]

*Figure 1: Overview map of study domain, depicting the marginal seas and schematic currents discussed in the text. Colourshading indicates depth. Marginal seas, islands, continents and schematic currents that are mentioned in the text are also marked. SCS: South China Sea, NEC: North Equatorial Current, SEC: South Equatorial Current, NECC: North Equatorial Counter Current.*

2. **Some of the markings on the figures are difficult to see, e.g. the labels and contours in Fig. 1, current vectors in Fig. 6. Switching grid colors to black and others to white may be one way to remedy this.**

As described above, we included an overview map as a new figure. Regarding Fig. 6, we decreased the number of arrows shown and increased their size.

3. **There are some repeated panels between figures. Perhaps some of the redundancy can be removed (though admittedly, I don't see an immediately obvious way). If not, please clearly mark where repeated.**

Yes, the colorshading in panels a) and d) of Fig. 4 are repeated in the same panels of Fig. 6. We agree that this is not strictly necessary. However, both figures support very different lines of reasoning and it makes it easier for the reader to follow our arguments to have all necessary charts in one figure. Combining figures 4 and 6 into a single figure seems unreasonable as it would result in a multipage figure. We therefore decided to repeat these two panels for the benefit of the reader. As you pointed out, we failed to mention the repeated figure and included a remark in the caption of Fig. 6.

4. **As you indicate they are not the same, what time step do the coarse and nested grid run at?**

The base model and the nest were integrated with time steps of 15 and 5 minutes respectively. We included this in the model description.

*L115-117: The base model provided boundary conditions for the nest at every time step of the nest (the base model and the nest were integrated with a time step of 5 and 15 minutes, respectively).*

5. **I'm glad you talk about the uncertainty in the atmospheric forcing product at the end. Perhaps you could add a justification as to why JRA55-do was chosen relative to another reanalysis product?**

Unlike other reanalysis products, JRA55-do is specifically designed to be used as an atmospheric forcing for OGCMs. It is of course based on an atmospheric reanalysis, but its surface fields are adjusted with respect to observations to resemble reality as close as possible. It also aims to provide closed heat and freshwater budgets (with respect to a given set of bulk formulas) to avoid model drift. We therefore favoured JRA55-do over other reanalysis products. We also acknowledge that this choice is somewhat arbitrary. We extended the model description to include these points.

*L102-105: The atmospheric forcing for all exper-iments is JRA55-do v1.3 (Tsujino et al., 2018) which builds on the JRA-55 reanalysis product but is specifically designedto be used as a forcing dataset for OGCMs. Most importantly, surface fields are corrected towards observations and heat andfreshwater-fluxes are balanced with respect to a defined set of bulk formulas.*

6. **I agree that the linear separation of WIND and BUOY is justified on the spatiotemporal scales you are looking at, but a further note clarifying what scales you**

**expect this to break down would be useful to a more general readership (which this manuscript attracts, through the implications for sea-level projections).**

The manuscript is concerned with interannual to decadal variability where we assume the linear separation to be valid. We do not suppress variability on timescales shorter than that so that the assumption is not valid on timescales shorter than one year. Furthermore, we would expect non-linearities to be of increasing importance with decreasing spatial scales and that the contribution of non-linear dynamics is not negligible for the mesoscale and below. We add a sentence to this respect to the methods section.

The length of our experiments prevents us from extending our analysis beyond the decadal to multidecadal scale. To which degree a quasi-linear superposition of wind- and buoyancy-driven signals remains approximately valid on longer timescales, must remain speculative: for example, wind-induced trends in SST could have increasing ramifications for surface heat fluxes, and effects of non-linearities could lead to amplification of small initial errors over the course of multi-century integrations, eventually obscuring the interpretation of individual sensitivity experiments.

*L137-137: Because we only suppress variability on interannual and longer timescales,this assumption is not valid for variability on shorter timescales.*

7. **Along these lines, I find the CLIM experiment quite interesting, and a nice way to capture the remaining variability. What's not clear to me is the relative role of the seasonal cycle vs nonlinear interactions in 'intrinsic variability. The discussion of eddy kinetic energy seems useful to this end, and I wonder if it could be quantified in the important regions (e.g., the central South China Sea). I freely admit that this discussion is outside my area of expertise, so apologies if this is not a well-posed comment.**

We would like to point out that the CLIM experiment does include a (repeated) seasonal cycle to avoid a possible misunderstanding. The seasonal cycle drives oceanic variability, and some of it might be subject to an upscale energy transfer and energize variability on longer timescales. Independent of this is the generation of intrinsic variability in CLIM related to non-linear ocean dynamics (e.g., the energy cascade of quasi-geostrophic turbulence). However, we can not quantify the individual contributions. Even the EKE, if based on the usual definition, includes both the contributions from forced intra-seasonal variations and dynamic instability processes, and hence appears of limited value for rigorously identifying their relative roles. Apart from that, the EKE (especially in the more energetic areas) represents a useful measure, and we already included this in our discussion about possible connections between the SCS and the Kuroshio boundary current region. We are unsure how our analysis could benefit further from a quantification of EKE in the SCS. Please feel free to clarify your remark.

8. **Can you elaborate on the methodology around line 175? Why did you not just calculate the variability contribution from BUOY? If you do that (assuming you can), does it compare well with the REF025-WIND contribution?**

Assuming, REF025 is a linear superposition of WIND and BUOY, both ways should yield the same result. This is not strictly true because all experiments include intrinsic variability. We therefore looked at the difference between REF025 and WIND rather than BUOY to estimate the effect of

buoyancy fluxes. When using BUOY directly, we find a much larger contribution of 24% which is likely due to intrinsic variability that is not negligible, in particular in the SCS (Fig. 3e, f).

*L199-200: When estimating the effect of buoyancy fluxes from BUOY directly, we obtain a much larger contribution of 24 %.The reason for this is intrinsic variability that is included in all experiments and adds to the variability in BUOY.*

9. **In your concluding sentence, you comment on the resolution of coupled circulation models. My initial thought would be that yes, coupling would indeed be important for sea-level projections as ENSO/PDO are subject to change, but that your findings would be largely robust. A comment to this effect would be good to see.**
   - **It would also be nice to wrap the paper up on a positive note, rather than the warning.**

[Figure]

We agree that a climate model would be more suited to analyse coupled ocean-atmosphere dynamics, and it would be interesting to repeat the regression analysis with output from a historical climate model run that provides sufficient resolution and compare the results. A possibility to generate high-resolution sea level projections without the need to run a climate model could be downscaling. Essentially, an uncoupled model is forced with the combination of the atmospheric long-term clime change signals (obtained from a climate model) and the high-frequency part of the historical forcing. Examples are Sun et al. 2012 or Feng et al. 2017. We used our last sentence to point to this approach.

*L376-378: Downscaled projections (e.g., Sun et al., 2012; Feng et al., 2017), in which an uncoupled high-resolution OGCM is forced with a long-term climate change signal obtained from a climate model, mightbe a way to address this issue.*

**Technical Corrections**
   - **When used as an adjective, e.g. "sea-level variability", sea level should be hyphenated (sea-level).**
Corrected
   - **In the first paragraph of the results, references to Fig. 1 are incorrectly marked as Fig. 3.**
Corrected
   - **On line 178, I think you mean sea-level variability (not just sea level).**
No, we refer to the positive trend between the early 90s and 2010 that constitutes a strong rise in sea level.
   - **L. 247, missing parenthesis before Fig 3.**

Corrected

We would like to sincerely thank the referee for the time and effort he or she put into this review and the helpful suggestions improving our manuscript. We address the issues raised by the referee below.

**I wonder if it is necessary to use the PDO index, as suggested by the authors, there are little differences in the result when just using the ENSO index. Maybe just use two frequency bands of the ENSO index.**

We agree that using two frequency bands of the ENSO index would be equally justified. We added another remark to the methodology that clarifies right from the beginning of the regression analysis that the PDO index, as it is used here, can be understood as the low-frequency component of the ENSO index.

*L211-213: Note that the PDO can be considered the low-frequency component of ENSO in this context to the extent that the PDO can be regarded as the low-frequency component of ENSO, an alternative, and largely equivalent separation could be build on the low-pass filtered Nino3.4 index instead of the PDO index.*

**As I understand both the ENSO and PDO indices are derived from the model results. Maybe it is necessary to show how the modeled indices compare with observations. It is not explained how the PDO index is derived.**

Agreed. We included indices based on observational data in Fig. A01 and also included a description of how both indices are derived.

*Footnote 6: The Niño3.4 index is defined as the area averaged sea surface temperature (SST) anomaly in the Niño3.4 region (5◦N–5◦S, 170◦W-120◦W) with respect to the monthly climatology and normalized by its SD.*

*Footnote 7: The PDO index is obtained via an Empirical Orthogonal Function of monthly SST anomalies in the Pacific north of 20◦N and defined as the leadingmode of variability. The climatological annual cycle is subtracted to remove long-term trends*

**Do different flavors of ENSO have an influence? Such as the so-called central Pacific ENSO.**

We did not test this but used the Nino34 index to capture both flavours of ENSO. Judging from SST regressions, the difference between Central Pacific and Eastern Pacific El Nino is most pronounced in the central to eastern Pacific (e.g. Ashok et al. 2007). Available observational data suggests that the impact in the region of interest is rather small. For example, Liu et al. 2015 showed that the ITF transport anomalies during both ENSO types are rather similar. We therefore suggest to not address this aspect in the manuscript.

*L268-271: Another question concerns the possibility of an asymmetric response to positive and negative ENSO phases. From an analysis of their individual contributions, we find that almost all regions with a non-negligible response to ENSO are characterized by similar amplitudes of SSH variability during both phases. Only the Arafura Sea responds stronger to the negative cycle (not shown).*

**Does the Indian Ocean have a decadal mode that could influence the region through oceanic Kelvin wave propagation or some teleconnection?**

Yes, there is evidence for Kelvin wave propagation from the Indian Ocean across the AMS into the Pacific (e.g. Yuan et al. 2013) and we considered to include an Indian Ocean mode in our analysis. The first choice would be the Indian Ocean Dipole Index (IOD), however the IOD does not show variability on the timescales considered here. The Indian Ocean is of course subject to decadal variability, but it appears mostly related to the Pacific or is even forced by Pacific climate modes. In particular in the southeastern tropical IO (see Han et al. 2014 for a review). We therefore consider it justified to not include an Indian Ocean mode in our analysis. We added a respective note to the manuscript.

*L214-216: The Indian Ocean Dipole Index (IOD; Saji et al. 1999) could account for possible impacts of the Indian Ocean. We do not include the IOD in our analysis because the index shows only very weak variability on the timescales considered here (not shown).*

**In Results, the first paragraph refers to Fig. 3, which in fact should be Fig. 1.**

Corrected

**Line 198: "off the northwest coast of Australia" - it is not clear what the authors refer to in the figure.**

It should read "off the northeast coast of Australia".

**Line 236: fix the sentence.**

Fixed

**Fig.7: the legend doesn't match the caption.**

We are not sure what you mean here. However, we changed the legend of Fig. 7 to show all lines. Please feel free to clarify this point.

[Figure]

*Figure 2: Linear regression of temperature and salinity to Niño3.4 (solid) and PDO index (dashed) from (a, b) reference experiments and (c,d) sensitivity experiments. All data has been filtered with an 8-year low-pass filter*

**Line 251: explain the meaning of the "advective nature".**

We rephrased the sentence it now reads:

*L286-287: The momentum flux experiments (WIND) reproduces the subsurface signals, highlighting the fact that they are due to wind-stress-driven advection, but shows no surface anomalies associated with ENSO and PDO cycles.*

We would like to sincerely thank the referee for the time and effort he or she put into this review and the helpful suggestions improving our manuscript. We address the issues raised by the referee below.

**General comments:**

1. **A study area map, with the name of the main places (which come up a lot during the text) would help to locate the reader. Just as presented in Figure 1 is not enough.**

We added an overview map of the region with additional labels as a new Fig. 1.

[Figure]

*Figure 3: Overview map of study domain, depicting the marginal seas and schematic currents discussed in the text. Colourshading indicates depth. Marginal seas, islands, continents and schematic currents that are mentioned in the text are also marked. SCS: South China Sea, NEC: North Equatorial Current, SEC: South Equatorial Current, NECC: North Equatorial Counter Current.*

2. **On line 89-91, the authors present the different nomenclatures for the region, and explain why they use 'Australasian Mediterranean Sea'. I must admit that I had never heard of such nomenclature, and found the word 'Mediterranean' in specific a bit puzzling at first. I wonder if using a more common nomenclature for the region, such as 'Tropical Asian Seas' and 'Southeastern Asian Sea', won't attract more readers.**

There is a broad variety of terms for the region but all lack precise definition. The term AMS seems to be used by a broader community and is also found in established textbooks (e.g. Tomczak and Godfrey, 2003). We also choose AMS in favour of other options because it includes the term "Mediterranean", which we consider an appropriate characterization of the region. Nevertheless, we agree that recent papers tend to use terms like "Tropical Asian Seas", "Southeastern Asian Sea" or "Maritime Continent" which is why we added the paragraph on the nomenclature in the first place.

We also acknowledge your argument that readers that are used to alternative terms might not be attracted by the term AMS. We therefore suggest including the terms "Tropical Asian Seas" not only in the introduction but also in the abstract.

*L2-5: Analyses of regional sea level in the densely populated regions of the neighbouring Australasian Mediterranean Sea (AMS; also called Tropical Asian Seas) are hindered by its complex topography and respective studies are sparse.*

3. **The results section is relatively long, and includes different things. I recommend the authors to create subsections to guide the reader better. For example, on Line 157 and L221 would be good places to start subsections.**

Agreed, we split the results section into three subsections.
*L178: 3.1 Impact of buoyancy fluxes on SSH*
*L204: 3.2 Linear regression of SSH on ENSO and PDO indices*
*L252: 3.3 Decomposition of SSH variability into thermosteric and halosteric contributions*

4. **Section 4 (Summary and conclusion), is more of a discussion section. The main findings are very clear in the abstract, but not so much in Section 4 (which is very long for a summary/conclusion). Some re-structuring of this Section would improve the text.**

Thank you for pointing this out. We renamed the section into "Summary and discussion" and restructured it into two subsections.

5. **The final 'general' conclusion of the paper is about the models' resolutions (from L332). However, while the effect of the resolution is clear for the temperature and salinity, it didn't seem to me so strong on the SSH variability itself. I don't think that these results have enough evidence to back the statements on L335,336.**

You are right that there are only minor differences with respect to SSH and we added this point to the last paragraph. However, we think that our results regarding temperature and salinity biases and the fact that many climate models are unable to resolve dynamically relevant straits and passages (e.g. Makassar strait or Mindoro-Sibuto strait) warrants our rather general warning. Additionally, a recent review on coupled ocean-atmosphere modelling over the region (Xue et al. 2020) support this with respect to ocean dynamics. We included the reference in the text.

*L368-376: A final point concerns the impact of model resolution in studies of variability patterns in this region. Although in our study we found only minor resolution-depended differences between the 1/4º and 1/20º-configurations with respect to SSH signals, the stronger biases in the temperature and salinity distributions seen in the coarser simulation should be taken as a reminder that a realistic representation of the dynamics of the AMS area with its complex, small-scale bathymetry can require a very high model resolution. The importance of resolving the narrow straits and bathymetric details for capturing the ocean dynamics inthe AMS was recently stressed in the review by Xue et al. (2020) of coupled ocean-atmosphere modelling for this region. In this context, it should be noted that the "coarse" 1/4°-grid used in this study would rather be in the higher-resolution category of current climate simulations with coupled ocean-atmosphere models,*

*suggesting that projections of regional trend patterns in this area need to be interpreted with due caution.*

6. **The observational dataset and variables used should be presented before the Results Section. Furthermore, it's important to mention that you use altimetry observations to validate the model, and that they are only available from 1993 (in contrast with your model that runs since 1958). Also how was the thermosteric and halosteric components computed? Or are they output of the model? This should be clarified. Also, where did the ENSO and PDO indices come from (from figure A1 it seems you computed it based on the model results, so would be good to say how you computed them. And does it match with the observations indices?)**

We computed the steric sea-level changes from the stored model output. We added another subsection on the diagnostics to the methods section. Another additional subsections now describes the observational data. Both climate indices were computed from the model output and compare well with observational estimates. We included a description regarding the computation and also added observations to figure A01.

*L142-147:*
*2.2 Observational datasets*
*We used two observational datasets to validate our model results. Satellite altimetry data is provided as a gridded product bythe Copernicus Marine Environment Monitoring Service (CMEMS) with a resolution of 1/4º that is available since 1993 (in contrast to our model simulations that start in 1958). Observations of temperature and salinity were taken from the World Ocean Atlas 18 (WOA18; Locarnini et al. 2019; Zweng et al. 2019) that is also available with 1/4º horizontal resolution and covers the period from 1958 to 2017.*

*L148-168:*
*2.3 Steric sea level*
*Sea-level changes are either due to changes of the total mass of the water column or due to density changes of seawater. The latter is referred to as steric sea level. Because seawater density is affected by temperature and salinity, steric sea level and can be separated into thermosteric and halosteric components, respectively. We diagnosed steric sea level changes from the stored model output:*

$$\frac{\partial \eta^{steric}}{\partial t} = -\int_{-H}^{\eta} \frac{1}{\rho}\frac{\partial \rho}{\partial t}\, dz \quad (1)$$

*where (η) denotes sea level, ρ is seawater density and H is ocean depth. Thermosteric sea-level changes are given by:*

$$\frac{\partial \eta^{thermosteric}}{\partial t} = \int_{-H}^{\eta} \left(\alpha \frac{\partial \Theta}{\partial t}\right) dz \quad , (2)$$

*where α is the thermal expansion coefficient and Θ is potential temperature. In the same way, halosteric sea level variationscan be expressed:*

$$\frac{\partial \eta^{halosteric}}{\partial t} = \int_{-H}^{\eta} (\beta \frac{\partial S}{\partial t}) dz \quad , (3)$$

*where β is the haline contraction coefficient and S is salinity.*

*Footnote 6: The Niño3.4 index is defined as the area averaged sea surface temperature (SST) anomaly in the Niño3.4 region (5◦N–5◦S, 170◦W-120◦W) with respect to the monthly climatology and normalized by its SD.*

*Footnote 7: The PDO index is obtained via an Empirical Orthogonal Function of monthly SST anomalies in the Pacific north of 20◦N and defined as the leadingmode of variability. The climatological annual cycle is subtracted to remove long-term trends*

7. **Did you do any analysis looking at 'specific' ENSO and PDO events? Does the explained variability change regarding an El Niño instead of a La Niña? Or does the position of the ENSO have an influence on your results? Maybe such analysis would enrich the paper even more.**

Regarding the symmetry between El Niño and La Niña:

We agree that this is a valid point. We did a composite analysis to check for differences between El Niño and La Niña events. We found the response to be symmetric with only minor difference in most seas except for two regions (see Figure below). Both, El Niño and La Niña trigger positive sea level anomalies in the western coastal regions of the SCS, which is not representative for the rest of the region. The total sea-level variability is also very weak in that area (see Fig. 1 in the manuscript). The second region is the Arafura Sea where La Niña produces much stronger anomalies than El Niño. However, we did not investigate this further but included a sentence to this end in the manuscript.

Regarding different ENSO flavours:

We did not test this but used the Nino34 index to capture both flavours of ENSO. Judging from SST regressions, the difference between Central Pacific and Easter Pacific El Ninos is most pronounced in the central to eastern Pacific (Ashok et al. 2007) and available observational data suggests that the impact in the region of interest is rather small. For example, Liu et al. 2015 showed the ITF transport anomalies during both ENSO types are rather similar. We therefore suggest to not address this aspect in the manuscript.

[Figure]

*Figure 4: Composites of SSH anomlies during (a) El Nino and (b) La Nina events and (c) the sum of both composites.*

*L268-271: Another question concerns the possibility of an asymmetric response to positive and negative ENSO phases. From an analysis of their individual contributions, we find that almost all regions with a non-negligible response to ENSO are characterized by similar amplitudes of SSH variability during both phases. Only the Arafura Sea responds stronger to the negative cycle (not shown).*

**Technical comments:**

1. **Sea level should be hyphened when used as an adjective (sea-level change Vs. change in sea level).**

Corrected

2. **Verbal times: In the introduction the authors use present (''the objectives are " (L75)); they use past tense in Section 2 ('we used a global ocean model" (L97)); future tense when presenting what is showed in the results ("First we will compare" (L139)); and back to present tense in Section 4. I recommend the authors to either keep it constant (i.e., using always present or past tense), or change the tenses accordingly to the Section, but keep it constant throughout the Section (i.e., present in the introduction, past in the methods (since it's something they already did), and present in the results and discussion, for example).**

Corrected

3. **The terms 'i.e.' and 'e.g.' are usually followed by a comma.**

This seems to be handled differently by different style guides. American guides tend to suggest the comma while British styles tend to omit it. We followed the British style and therefore don't put a comma beyond both abbreviations.

4. **Use of one-sentence paragraphs (for example, L258-259, L289-290) or very short paragraphs (e.g., L71-73, L84-86) should be avoided.**

Corrected

**Specific comments**

**1. ENSO and PDO are not defined (neither in the abstract or the main text). Although they are well-known acronyms, it should be defined.**

Corrected

**2. Whenever talking about sea-level trends, it is important to mention the period that it referred to. For example, L24 a trend is given, but no period. Also the first sentence of the paper is 'over the last three decades', but then the older reference is 2013, so I assume you mean 1980-2010? Not including 2010-2020? Adding the period to the sentence would clarify this.**

Clarified

**3. L22 you mention that some areas in the western tropical pacific have received a lot of attention. Maybe adding a more specific example would be nice.**

We refer to the "western tropical Pacific" and don't see where specific subregions are mentioned. Examples for studies are given after the first sentence of the paragraph.

**4. L34: You mention the 'Luzon Strait'. I have no idea where it is. Having a study are map, with such names would help.**

Please see response to your general comment No. 1.

**5. L35 'might amplify the signal'. Which signal?**

We refer to the ENSO-related SSH anomalies mentioned in the preceding sentence. We rephrased the sentence accordingly.

*L36-37: Local wind stress curl, also related to ENSO, might amplify these sea-level anomalies in the central SCS (Cheng et al., 2016).*

**6. L43-44: is the ITF variability governed large-scale climate events (like ENSO and PDO), or by the climate indices (like Niño3.4)?**

We changed the sentence to:

*L44-45: The ITF variability on interannual time scales is mostly a response to variations of the Pacific trade winds and governed by large-scale climate modes.*

**7. L52: Wijffels and Meyers showed (instead of 'could show'), based on XBT observations, that sea-level anomalies... Would also be good to mention which period of observations they used.**

L53: *1984-2001*

**8. L57: You mention 'mass and steric components'. I think it would be good to define to the reader what exactly are these components (not everyone will know). And mainly**

**the steric contribution could be better explained, as you discuss it further on your results.**

Please see our reply to your general comment No. 6.

**9. L61: are you talking about sea-level trends? ('reconstructed decadal trends during the...'). Make it clear to the reader.**

Yes, sea-level trends. Clarified.

**10. The term halosteric appears for the first time on Line 67, and then again with the term thermosteric in the objectives of the paper (Line 83). But you didn't introduce what thermo and halosteric changes are. Although the readers of Ocean Sciences might be familiar with it, in my opinion you could briefly define thermo and halosteric changes before having it as one of the objectives of the paper.**

We rephrased the third objective to avoid the term "halosteric" and introduced it in the following section.

*L85: To analyse the individual contributions of temperature and salinity changes to steric sea-level variability.*

**11. Line 75 should be connected to the previous paragraph.**

Corrected

**12. Line 104 should be connected to the previous paragraph.**

Corrected

**13. Lines 104-106: Won't this freshwater budget correction affect then your halosteric analysis?**

Yes, this could affect halosteric sea level. We have the freshwater flux of both corrections available and find their variability in the region at least an order of magnitude smaller than the net freshwater flux. Even in the coarse resolution experiments that employ a stronger restoring than the high resolution experiment. We therefore assume the effect on halosteric sea level to be very small.

**14. L122: is there a specific reason for using the years 1990-1991 as your climatological forcing?**

Yes, that is the period recommend by Stewart et al. 2020. We clarified this in the text.

*L127-129: More specifically, we extracted the recommended 12-month subset from May 1990 to April 1991 from the full forcing dataset and applied its individual fields repeatedly to suppress the interannual variability entirely or only for the computation of momentum or buoyancy fluxes, respectively.*

**15. L129: Add 'respectively' at the end of the sentence: 'isolate the momentum forced and buoyancy forced variability, respectively'.**

Corrected

**16. L134 should be connected to the previous paragraph.**

Corrected

**17. L139: Here you start about your validation with observations. It's important to highlight that you are using satellite altimetry observations, and that they only start in 1993**

Please see our reply to your general comment No. 6.

**18. L143: Here and in the rest of the paragraph I think you meant figure 1?**

Yes, corrected

**19. L149: biases in relation to what? Model biases? Make it clearer**

Clarified

*L170-171: Figure 3 shows subsurface maps of mean fields and model biases of temperature and salinity with respect to observations from WOA18, averaged over the upper 400 m, from both hindcast simulations.*

**20. L151: What is the reference period of the mean fields? Is it the same as the model biases (which I am assuming is for the entire period?)**

WOA18 Data is available from 1955 to 2017 and the mean field show a long-term average over this period. Model biases are computed over the full integration period of the models (1958-2016). Details are added to the figure caption.

*Figure 3. Mean fields (1955–2017; upper row) and model biases (1958–2016; middle and bottom row) averaged over the upper 400 m for (a-c) temperature and (d-f) salinity. Note that data is only shown where the water depth exceeds 400 m.*

**21. Figure 2: Here locations where you don't have data (which is a lot of the AMS actually) has been plotted with the same color as land, which was confusing. So I suggest you to plot it in a different color. I was also left wondering why you don't have data there (I can understand from the observations, if are areas that are too shallow... but I would expect the model to have values there as well). So it might be worth adding one sentence about it in the main text and/or in the caption of the figure.**

The figure shows an average over the upper 400 m and data is only shown where the water is at least 400 m deep. All other areas are marked as land. We clarified this in the caption.

*Figure 3. [...] Note that data is only shown where the water depth exceeds 400 m.*

**22. L162: I found the word 'confirming' a bit of a bold statement, I suggest replacing it with 'indicating'.**

Changed "confirming" to "indicating".

**23. L164 and Figure 3: Here you sat that you are using different colorbar for the same figure. I can understand why you want to do this, since if you keep the range of panel d and e to 0-4 we won't be able to see the patterns... But I still find it a bit misleading. Even though the colorbars are indicating that they range only up to 1, I find it harder to compare the panels, which is a bit the point of the plot. So it's up to you, but I would suggest fixing the colorbars throughout the figure.**

We agree that the different colourbars can be misleading. However, as you state correctly, using the same colourbars in all panels makes it impossible to distinguish any patterns in panels d) and e). In particular, the pattern in panel e) is an important result. We deliberately choose the ranges of 4 and 1 to facilitate the comparison (same colour means difference by a factor 4). We therefore suggest keeping the different colourbars.

**24. Both paragraphs at L167-179 are just talking about REF025 experiment. What about REF005?**

The paragraphs are on the sensitivity experiments that are only available for the REF025 experiment.

**25. L185: 'e.g.' should be followed by a comma. And the first name of the authors of the reference are appearing (Thomson, R.E.)**

Please see our reply to your technical comment No. 3 regarding the comma. We fixed the reference.

**26. L190; Bring the reference to Figure 4 here ('REF020 and REF005 show a similar response to positive ENSO cycles (Figure 4.a,b)'.**

Changed

**27. Figure 3: Why are the grid lines here marking only 10˚N, 0˚ and 10˚S?  And the longitudes are on a different spacing than in Figure 1 and 2. This is very small detail, but you could use always the same gridline spacing.**

Changed

**28. L200: Bring the reference to Figure 4d.e. here: 'the linear response to the PDO index (Figure 4.d.e.)...'**

Changed

**29. L200: I think you meant a 'and' here instead of 'to': 'strong amplitudes of 3 and 4 cm'.**

Indeed

**30. L212: Figure 5d is about PDO... I thought you were talking about ENSO, so figure 5c?**

Changed

**31. L2017: 'PDO does not drive any buoyancy flux driven variability', in the AMS right? Because I can seem some variability on the surrounding seas around the AMS.**

Changed accordingly.

*L248: The PDO does not drive any buoyancy-flux-driven SSH variability in the AMS (Fig. 6 f).*

**32. L224: How do you know that the mass fluctuations are small? Is this based on previous research, if so, then reference it.**

This statement is based on our own research. But the claim is also in line with previous research. We added a reference.

*L254-256:We neglect sea-level anomalies due to mass fluctuations because their relative contributions are small everywhere except on shelf regions (not shown; e.g. Forget and Ponte 2015) where the total variability is weakest.*

**33. Figure 5: Add variable name to the colorbar (R-squared), and also mention in the caption which coefficient you are showing (R-squared)**

Corrected

**34. L233: halosteric anomalies complement (not amplify) the thermosteric signal (it is not because of the halosteric variation that the thermosteric signal will be higher, but you will have a total steric change that is higher).**

Changed "amplify" to "complement".

**35. L245: Remove 'does': changes during PDO cycles also manifest in the vertical profiles'.**

Corrected

**36. Figure 6 caption: linear regression with (instead of to); move the 'of upper ocean currents' to the end of the sentence or to another sentence (right now it seems you made the correlation with the currents, but I believe you are just showing the currents). And are those the mean current velocity? Make it clearer.**

The figure shows a regression of currents and SSH with ENSO and PDO index.

**37. First paragraph of 'Summary and conclusion' should be connected to the second one.**

Corrected

**38. L269: pressure gradient between the Pacific and (not in) the Indian Ocean.**

Corrected

**39. L270: Remove 'again'.**

Corrected

**40. Line 289 should be connected to previous paragraph.**

Corrected

**41. L306: Add 'current' with Kuroshio here and in the following mentions of the current (Kuroshio current flows...)**

Corrected

**42. L324: 'McGregor demonstrated (not could demonstrate).**

Corrected

**43. General conclusion statement: The resolution effect is clear for the biases, but it doesn't seem to have had a significant impact in your SSH variability analysis. I'm not sure if your final and general conclusion should be about this.**

Please see our reply to your general comment No. 5.

**References**

Ashok, K., Behera, S. K., Rao, S. A., Weng, H., and Yamagata, T. (2007), El Niño Modoki and its possible teleconnection, *J. Geophys. Res.*, 112, C11007, doi:10.1029/2006JC003798.

Feng, M., Zhang, X., Sloyan, B., and Chamberlain, M. (2017), Contribution of the deep ocean to the centennial changes of the Indonesian Throughflow, *Geophys. Res. Lett.*, 44, 2859– 2867, doi:10.1002/2017GL072577.

Han, W., Vialard, J., McPhaden, M. J., Lee, T., Masumoto, Y., Feng, M., & de Ruijter, W. P. (2014). Indian Ocean Decadal Variability: A Review, *Bulletin of the American Meteorological Society*, *95*(11), 1679-1703.

Liu, Q.-Y., Feng, M., Wang, D., and Wijffels, S. (2015), Interannual variability of the Indonesian Throughflow transport: A revisit based on 30 year expendable bathythermograph data, *J. Geophys. Res. Oceans*, 120, 8270– 8282, doi:10.1002/2015JC011351.

Sun, C., Feng, M., Matear, R. J., Chamberlain, M. A., Craig, P., Ridgway, K. R., & Schiller, A. (2012). Marine Downscaling of a Future Climate Scenario for Australian Boundary Currents, *Journal of Climate*, *25*(8), 2947-2962.

Tomczak, M. and Godfrey, J. S.: Regional Oceanography: an Introduction, Daya Publishing House, Delhi, 2 edn., 2003.

Xue, P., Malanotte-Rizzoli, P., Wei, J., & Eltahir, E. (2020). Coupled Ocean-Atmosphere Modeling Over the Maritime Continent: A Review. Journal of Geophysical Research: Oceans, 125(6). http://doi.org/10.1029/2019JC014978